# MXene-Based Materials for Solar Cell Applications

**DOI:** 10.3390/nano11123170

**Published:** 2021-11-23

**Authors:** Zhe Shi, Rasoul Khaledialidusti, Massoud Malaki, Han Zhang

**Affiliations:** 1School of Physics and New Energy, Xuzhou University of Technology, Xuzhou 221018, China; laser532@126.com; 2Department of Mechanical and Industrial Engineering, Norwegian University of Science and Technology (NTNU), 7491 Trondheim, Norway; rasool.khaledi@gmail.com; 3Department of Mechanical Engineering, Isfahan University of Technology, Isfahan 84156-83111, Iran; 4Shenzhen Engineering Laboratory of Phosphorene and Optoelectronics, Collaborative Innovation Center for Optoelectronic Science and Technology, College of Optoelectronic Engineering, Shenzhen University, Shenzhen 518060, China

**Keywords:** MXene, solar cell, 2D material, energy, environment

## Abstract

MXenes are a class of two-dimensional nanomaterials with exceptional tailor-made properties, making them promising candidates for a wide variety of critical applications from energy systems, optics, electromagnetic interference shielding to those advanced sensors, and medical devices. Owing to its mechano-ceramic nature, MXenes have superior thermal, mechanical, and electrical properties. Recently, MXene-based materials are being extensively explored for solar cell applications wherein materials with superior sustainability, performance, and efficiency have been developed in demand to reduce the manufacturing cost of the present solar cell materials as well as enhance the productivity, efficiency, and performance of the MXene-based materials for solar energy harvesting. It is aimed in this review to study those MXenes employed in solar technologies, and in terms of the layout of the current paper, those 2D materials candidates used in solar cell applications are briefly reviewed and discussed, and then the fabrication methods are introduced. The key synthesis methods of MXenes, as well as the electrical, optical, and thermoelectric properties, are explained before those research efforts studying MXenes in solar cell materials are comprehensively discussed. It is believed that the use of MXene in solar technologies is in its infancy stage and many research efforts are yet to be performed on the current pitfalls to fill the existing voids.

## 1. Introduction

The sun is one of the most significant sustainable energy resources, and harnessing as much energy as possible has always been demanded, as the other energy sources, such as fossil fuels, are limited. One hour of the sun’s power to the Earth could satisfy global energy demands for one year; however, the most critical drawback of common solar systems has been the relatively high costs. Initial costs, such as panels, inverters, batteries, and other installation expenses, are yet to be addressed. Further, gathering solar energy is enormously weather-dependent, and the more energy you need to produce, the more solar panels with enough space are required [1,2,3].

Solar energy systems have a wide variety of applications with low maintenance costs ranging from satellites in space to water distillation processes. Thanks to technology and novel materials, solar systems have drastically been improved, turning solar energy systems into a clear and efficient energy source. With the clean energy trend gaining momentum, investigators are intensely looking at ways to harness as much energy as possible from the sun for a wide variety of commercial applications. The first-generation solar cell made from crystalline silicon is the most common and highly efficient solar cell; however, the manufacturing and maintenance expenses associated with this type are considerable. The second-generation solar cells based on thin films are cost-effective but not efficient enough compared to the other generations. The third-generation dye-synthesized, perovskites, and organic solar cell systems aim to develop high-performance, inexpensive materials with enhanced performance. The third-generation solar cells are going to make way for the breakthrough, cost-effective solar materials using nanomaterials. Replacing conventional materials with those novel 2D nanosheets with high aspect ratios and electrical conductivity would be a fundamental breakthrough in solar cell technologies. As a family of recently discovered 2D materials, MXene is a few-thick-atoms layer of transition metal carbides, nitrides, and carbonitrides with excellent thermal, electrical, and ion diffusion behaviors [4].

In terms of the layout of the present review, those candidates for solar cell applications are discussed first, and then the fabrication routes of various types of solar cells are elaborated. A detailed discussion on the synthesis and key properties will then be introduced. Those research efforts on MXenes in solar cell applications have comprehensively and critically been reviewed and discussed to fill the existing voids for future research.

## 2. Nanomaterial Candidates for Solar Cell Applications

The unusual optical and electronic properties of 2D materials have brought excitement to the solar community, which has led them to be widely investigated for next-generation solar cells and other optoelectronic devices. Several 2D semiconductor materials such as graphene, perovskite, transition metal dichalcogenides, graphdiyne, and black phosphorus have been realized for solar cell applications. Transition metal carbides, carbonitrides, and nitrides (MXenes) are the latest additions to the 2D material families with photonics and optoelectronics applications.

Graphene with zero-bandgap semi-metal and ambipolar electrical characteristics exposes well-intentioned optical properties with high transparency (97.7% transmittance in the visible spectrum) and outstanding electrical properties with high electrical conductivity (≈10^4^ Ω^−1^ cm^−1^). Beyond graphene, some other 2D materials can also be served in solar cell technologies. For example, the excellent mechanical properties of transition-metal dichalcogenides and black phosphorous make them ideal candidates for flexible and stretchable next-generation electronic devices. Depending on the nature of the elements and surface terminations, MXenes exhibit a semiconductor-like behavior with a direct bandgap at the monolayer that can be used as an active layer in ultrathin flexible solar cells. These materials have been glorified owing to a unique combination of their semiconducting behavior, mechanical properties (i.e., high tensile strength and Young’s modulus), and ultralight weight properties making them a promising candidate for optical devices [1,2,3,4]. Herein, a comprehensive review of the present state-of-the-art of 2D-materials for solar cell applications is presented to employ the recent advances of 2D materials for solar cell technologies.

### 2.1. Perovskite

Perovskites are a large family of materials with the same crystal structure and chemical formula, which is named after the mineral with that structure. The general chemical formula is ABX_3_, wherein A and B represent two positively charged metal atoms of very different sizes; X represents a negatively charged atom bonded to A and B, as depicted in Figure 1. The radii of A and B as cations are generally between 1.6 Å and 2.5 Å [5]; the radii of B atoms are smaller than A, forming a cube around the A atom. Negatively charged atoms are chemically bonded to every side and the mid-way through the cube of B atoms [6,7,8].

The term ‘perovskite solar cell’ is originated from the perovskite crystal structure of the absorber materials with ABX_3_ structure. Methylammonium lead trihalide with the chemical formula of CH_3_NH_3_PbX_3_ is one of the most promising perovskite absorbers, where X = I, Br, and/or Cl, with the optical bandgap in a range of 1.57–2.3 eV, depending on the X atoms. Shown in Figure 1b, the central A site of the perovskite crystal structure is filled with Methylammonium cation (${\mathrm{CH}}_{3}{\mathrm{NH}}_{3}^{+}$), surrounded by 12 nearest-neighbor X ions in corner-sharing PbX_6_ octahedra. Another example of a promising perovskite absorber is Formamidinium lead trihalide with the chemical formula of H_2_NCHNH_2_PbX_3_ with the bandgaps between 1.48 and 2.2 eV. Since the minimum bandgap of Formamidinium lead trihalide (i.e., 1.48 eV) is closer than that of Methylammonium lead trihalide (i.e., 1.57 eV) to the optimum value for a single-junction cell, it may have the capability of higher efficiencies [9]. The perovskite crystal structure of these two materials, the atomic structure of methylammonium and formamidinium as A-site cations, and the absorbance spectra of these materials are illustrated in Figure 2.

As promising candidates for solar cell applications, perovskite materials have several exclusive features: (i) the low cost of the raw materials and the fabrication methods; (ii) being light-weight, thin, and flexible; (iii) the high absorption coefficient, e.g., absorption of the whole visible solar spectrum by ultrathin films of ~500 nm thick [10,11]. The large-scale manufacture of perovskite solar cells could therefore afford a cheap, versatile source of clean and renewable energy. Recently, perovskite solar cells have significantly been improved so that their conversion energy is enhanced from almost 3.0% in 2006 to about 25% in single-junction architectures and about 29% in silicon-based tandem cells in 2020 [12]. On the other hand, the perovskite solar cells suffer from the following issues:The long-term instability owing to the degradation pathways, including the external factors (i.e., water, light, and oxygen) and also degradation upon heating [13,14]. The stability could be improved by applying some approaches such as (a) changing the components and including inorganic cations such as rubidium or cesium to make the mixed-cation structures [15], (b) providing more hydrophobic conditions for the UV-stable interfacial layers to improve the stability, for example, using SnO_2_ instead of TiO_2_ being vulnerable to UV degradation [16], and (c) improving the surface passivation by linking 2D layered perovskites with conventional 3D perovskites [17,18].The toxicity associated with the use of lead in perovskite solar cells. Although the amount of lead content used in the solar cells is much smaller than those used in other technologies such as lead-based batteries, it has been informed that the existence of lead in the large-scale implementation of perovskite solar cells could be problematic [19]. It has also been revealed that the toxicity of lead in the solar cell is negligible, and it is much less than other materials (e.g., cathode) in the cell [20]. There is a possibility to remove the toxicity of lead by the replacement by tin in perovskite solar cells, but it may weaken the power conversion efficiency [21]. Of note, it has been found that tin may have a higher toxicity impact than lead in the perovskite solar cells; hence it is required to replace lead with the other elements of lower toxicity [22].The current-voltage hysteresis is normally realized in the devices. The important parameters affecting the hysteresis are still in question; however, the mobile ion migration in conjunction with the high levels of recombination could be introduced as the most important parameter [23]. To reduce hysteresis, those strategies such as varying the surface passivation and cell architecture, increasing lead iodide content, and reducing the recombination could be followed [24].

### 2.2. Transitional Metal Dichalcogenides

2D transition metal dichalcogenides (TMDs) are a class of materials with the general formula of MX_2_, wherein M represents an early transition metal (i.e., Mo, W, Ti, and the like) and X represents a chalcogen (i.e., S, Se, Te). As shown in Figure 3a, the family of TMDs materials consists of 16 transition metals and three chalcogen atoms. It is worth noting that only a few TMDs were built with Co, Rh, Ir, and Ni, such as NiTe_2_ [25]. Some examples of this family include MoS_2_, WS_2_, and MoTe_2_, being abundantly accessible. In a 2D TMD monolayer structure, a single atomic layer of the transition metal (e.g., Mo) is sandwiched between the two atomic layers of chalcogen (e.g., S), where transition metal atoms are covalently bonded with chalcogens. Due to weak van der Waals interactions between the two adjacent MX_2_ slabs, TMDs can be exfoliated to single layers [26]. The MX_2_ monolayer is 6–7 Å; for example, a monolayer MoS_2_ is only 6.5 Å thick [25,27]. 

According to literature, the electronic properties of the family of 2D TMDs range from metallic to semiconducting materials, e.g., while VSe_2_ and NbS_2_ are metals, WTe_2_ and TiSe_2_ are semimetals, and MoS_2_, MoSe_2_, WS_2_, and WSe_2_ are semiconductors [25,27,32]. Indeed, the filling of d orbitals of the transition metals in a TMD has a major role in the electronic structure. Since the oxidation state of the transition metals and the chalcogens in MX_2_ compounds is respectively +4 and −2, the number of d orbitals electrons of the transition metals in TMDs varies from 0 to 6 from group 4 to 10 of the periodic table; that is why varied electronic properties can be understood for different TMDs, depending on the combinations of the transition metal and chalcogen elements. Similarly, many other phases in monolayer TMDs may be recognized. 

The monolayer TMDs, depending on the M and X elements, can be formed in two different stable phases of the trigonal prismatic or octahedral phases (see Figure 3). Conventionally known as the 1H phase (Figure 3c), the trigonal prismatic phase possesses a hexagonal symmetry with trigonal prismatic coordination of the metal atoms. The octahedral phase is typically classified as the 1T phase (Figure 3d), which has a tetragonal symmetry with octahedral coordination of the metal atoms. As stated, the atomic structure of the TMD layers and their electronic properties depends on the filling of the d orbitals of the transition metal. The TMDs exhibit semiconductor-like properties if the d orbitals of the metal are filled while partial filling produces metallic behavior [29]. The type of phases and the symmetry of the monolayer TMDs are also highly dependent on the filling of d orbitals. The group 4 elements (e.g., TiX_2_) and most of the group 6 elements (e.g., MoX_2_) have trigonal prismatic phases; the group 5 elements may have both trigonal prismatic or octahedral phases; the group 7 elements (e.g., MnX_2_) have a typically distorted octahedral structure (Figure 3d); the group 10 elements (e.g., NiX_2_) have an octahedral phase.

Here, we focus on the 2D TMDs semiconductors (Figure 3b,g). Generally, semiconductor TMDs are capable of adsorbing photons with energy equal to or larger than their bandgaps. The bandgap of most TMDs are dependent on their thickness and have similar common features (Figure 3f), the bandgap transfers from indirect-to-direct by reducing the thickness from multiple layers to monolayer owing to the strong quantum confinement effects [30]. Since semiconductors are more efficient as emitters when they possess the direct bandgap, monolayer semiconducting TMDs provide orders of magnitude improvement of the photoluminescence and have a higher emission efficiency than their multilayers; for example, the emission efficiency of monolayer TMDs is nearly 10^4^ greater than their bulk materials [33]; the bandgap of TMDs are in the range of 1.0 and 2.5 eV, which allows covering the spectra from the visible to infrared area (between 400 nm and 700 nm). TMDs may exhibit robust excitonic effects in the interaction with photons exposing several absorption peaks from ultraviolet to adjacent infrared spectra; 30% of which can be absorbed due to excitonic and interband transitions [33]. In addition, TMDs offer potential nonlinear optical applications (e.g., wavelength convertors) upon the broken inversion symmetry of the structure (typically odd layers of TMDs), providing a double output frequency when a laser excites the crystal [34,35]. Generally, in the monolayer TMDs with direct bandgap, the interband transition at near of the non-equivalent K points of the 2D hexagonal Brillouin zone, where both conduction and valence band edges are located, are coupled to right/left circular photon polarization states. These so-called valley-dependent optical selection rules can be the result of breaking the inversion symmetry of the structure; therefore, the broken inversion symmetry of TMDs can also induce valley selective optical properties owing to a strong spin-orbit coupling in TMDs, providing superior flexibility for optical modulation [36].

### 2.3. Graphene

As the thinnest 2D nanomaterial of ~0.34 nm thickness, graphene consisting of carbon atoms in a hexagonal honeycomb lattice on a 2D plane that the neighboring atoms being covalently bonded by three sigma bonds and one π-bon can be synthesized with varied sizes from several nanometers to a few centimeters (see Figure 4a). Graphene can be found in two different edge types (i.e., armchair and zigzag) that normally co-exist in graphene materials, while the properties of these two types are different. 

Exceptional properties of graphene have led them to be used in many applications such as energy storage and conversion, electronic devices, optoelectronic circuits, and biomedical appliances. Charge carriers in graphene are confined to one atomic layer thick, demonstrating considerable intrinsic charge mobility (120,000 cm^2^/Vs) at 240 K as well as exceptionally great velocities at high fields (4 × 10^7^ cm/s), distinguishing graphene from any other semiconductor [37]. Graphene possesses an exceptional specific surface area (2630 m^2^/g) [38], superior thermal conductivity (5000 W/mK) (even more than diamond and graphite) [39], a great Young’s modulus (1.0 TPa), intrinsic mechanical properties (130 GPa) due to the strength of covalent bonds between the carbon atoms [40], and superb optical transmittance (97.7%) in the visible region with very small reflectance (<0.1%), with the transparency that can be reduced by increasing the number of layers [41]. Since graphene has been realized as a low-cost, light-weight, high-performance, efficient, and flexible energy material, it has mainly been used in energy-efficient applications. As proof, graphene has been applied for (a) supercapacitors to support low-cost and high yield energy storage [42,43], (b) electrodes in lithium–ion batteries [44,45], (c) catalysts for advanced oxidation processes [46], and (d) the electrodes, and both the anode and the cathode in solar cells due to its high transparency and conductivity for producing highly efficient photovoltaic devices [47,48]. 

In terms of the electronic property, graphene is a semi-metallic, zero bandgap semiconductor because the conduction and valence bands meet at the Dirac points, in which the charge carrier may form linear electronic dispersion cones. In more detail, the conduction band (π* band) and valence band (π band) meet at six vertices in momentum space on the edge of the hexagonal Brillouin zone; four of these six points are equivalent because of the symmetry and only two of them are distinct (K and K’) (Figure 4b).

In view of the optical property, as stated, graphene has brought excitement to the solar cell community owing to its excellent transparency (97.7%) in the visible region with minor values of reflectance (<0.1%) [41]; therefore, the community started to employ graphene for solar cell applications. To date, different types of graphene-based solar cells have been reported; for example, graphene-based materials have been employed for organic solar cells (OSCs), dye-sensitized solar cells (DSSCs), and perovskite solar cells (PSCs) [48]. The use of graphene-based materials for OSCs has been confirmed as electrodes and charge acceptor materials (both hole-transport layer (HTL) and electron-transport layer (ETL)). As electrodes, graphene anode and cathode used in the flexible OSCs could provide power conversion efficiencies of 6.1% and 7.1%, respectively [51]. The proficiency of graphene application as electrodes on the flexible substrates facilitates the large-scale roll-to-roll process to produce OSCs [52,53].

Apart from the application as electrodes, graphene has been heavily investigated as the charge acceptor materials owing to their superior charge mobility introducing them as a suitable candidate for chemical modification and bandgap tuning [54]. A semiconducting polymer being a bi-continuous composite of an electron donor and an electron acceptor is responsible for absorbing the photons in the OSCs. The direct contact of the electrodes and electron donor-acceptor materials causes a decline in the current leakage required for functional layers at the interface to support charge extraction and collection; therefore, additional layers, so-called buffer layers or interfacial layers, are highly required to effectively pass or block specific types of charge (electron or hole). It has been proved that using the 2D conjugated structure of the graphene as a proficient interface with the donor could provide a faster exciton separation and charge transfer [55]. It has been revealed that graphene oxide (GO) is able to block the electrons and it is also extremely qualified as a hole-transport material [56,57]. Further, the application of graphene-based materials such as HTL, and their applications as ETL, have also been confirmed; the functionality of ETL is to develop the efficiency of cathode electrodes to collect and extract negative charge carriers [58]. It is required that the work function of the electron-extraction material be reasonably low to induce an electrical field through the active layer and transport electrons in the direction of the cathode. In this way, graphene oxide (GO) with the work function of 4.6 eV can directly be applied as ETL [59].

Graphene-based materials have also been applied as the solid-state electrolyte in dye-sensitized solar cells (DSSCs). Conventionally, liquid electrolytes are widely used as a transport medium for DSSCs due to high conductivity, low viscosity, easy preparation, and great conversion efficiency [60]; however, liquid electrolytes suffer from high-temperature instability, leakage, evaporation, and integration with flexible devices. To address these difficulties, solid-state electrolytes based on organic–inorganic hybrids and polymer electrolytes are developed [60]. The graphene-based materials have been used to make these solid-state electrolytes in DSSCs with a comparative efficiency compared to the conventional liquid electrolytes [61]. 

Finally, graphene-based materials have been applied in perovskite solar cells (PSCs). For example, a 3D scaffold of a graphene-based material may be employed as an interface layer between electron transfer and absorber layers in PSCs, providing an improvement in the device performance and perovskite structure stability. The aforesaid improvement is mainly attributed to the superior crystallization and favored orientation order of perovskite structure and improved energy level alignment at the perovskite interface [62,63].

### 2.4. Graphdiyne

As a member of 2D carbon materials, graphynes are one-atom-thick planar sheets formed by the mixed hybridization of sp- and sp^2^-bonded carbon atoms developed in the crystal lattice [62,63]. Owing to the mixed hybridization of carbon atoms, graphynes’ structures and properties are different from other carbon allotropes such as graphene, including pure sp^2^-hybridized carbon atoms [62,63]. The graphynes possess a 2D system similar to graphene comprising acetylenic linkages (sp components) in the place of carbon–carbon single bonds of graphene. The mixed hybridization of sp- and sp^2^-bonded carbon atoms in single-layer graphynes make covalent bonding between the atoms and provide many different crystal structures and symmetries. Indeed, “graphyne” is a general name of a category of systems, where the two neighboring sp^2^-hybridized carbon atoms are interlinked by *n* “–C≡C–” bonds and are classified and named based on the number of “–C≡C–” bonds. As shown in Figure 5, they are called graphyne, graphdiyne, and graphyne-*n*. The percentage of acetylenic linkages differs between different graphynes. Graphyne, for example, has a 100% percentage of acetylenic linkages, following the replacement of all the carbon–carbon bonds in graphene by acetylenic (“–C≡C–”) linkages; however, this percentage in graphdiyne is 66.67%, which is resulted from the replacement of two-third of carbon–carbon bonds in graphene by acetylenic (“–C≡C–”) linkages [64,65]. Although the structure of graphynes was proposed for the first time in 1987 [66], the first graphyne film (i.e., graphyne-2 or graphdiyne) was synthesized in 2010 because of its high formation energy and flexibility of sp-bonds [67]. Since then, a more significant number of the research papers related to this class of 2D materials concern the graphdiyne; here, a particular focus is paid on graphdiyne (GDY).

As the most stable graphyne containing diacetylene (“–C≡C–C≡C–”) interlinks, GDY materials possess a direct bandgap of 0.44–1.47 eV (different values are reported based on the calculation methods), and a high carrier electron mobility of about 10^5^ cm^2^/Vs at room temperature introduces GDY materials as a promising candidate for the applications of photocatalytic, water remediation, and energy storage [65,70,71,72,73,74]. The mechanical properties of GDY with a Young’s modulus of about 0.25 TPa are weaker than graphene owing to the existence of weak carbon–carbon single bonds in the acetylenic linkages [74]. Although GDY materials exhibit unique properties for different practical applications, there is still a gap between the theoretical results and the reality due to the difficulty of producing GDY as a perfect single-crystal structure containing sp- and sp^2^-hybridized carbon with a thickness of only one atom [65,75].

In view of the electrical property, a single-layer GDY is a semiconductor material with a narrow bandgap shown in Figure 5c with a bandgap of 0.46 eV at the *Γ* point [69]. The carrier mobility of GDY as a critical parameter for the electrical properties of semiconductor materials in the field of electronics is significant (electron mobility of 10^5^ cm^2^/Vs, being one order of magnitude higher than the hole mobility) at room temperature [65,70]. The bilayer and trilayer GDY possess different electronic properties than the single-layer one, depending on the stacking methods [76]. It has been reported that the bilayer and trilayer GDY exhibit a semiconducting behavior with the bandgap of about 0.35 eV and 0.33 eV, respectively, upon the stacking of the benzene ring in the Bernal mode introduced as the most stable stacking mode for bilayer and trilayer GDY [76]. Therefore, both bilayer and trilayer GDY display a narrower bandgap than the intrinsic bandgap of the single-layer GDY, while the direct bandgap is reserved. 

Although the electronic properties of GDY have extensively been investigated, optical properties are scarcely evaluated. Among these relatively infrequent studies, the optical absorption spectrum of GDY as the most critical optical properties has been evaluated theoretically and experimentally [65,75,77]. Although the theoretical and experimental results do not precisely match, they approve the same trend confirming the optical absorption spectrum with three peaks. It is considered that the first peak is originated from the transitions around the bandgap, and the two others are originated from the transitions around the Van Hove singularities at the *M* and *K* points. In addition, the adsorption range of GDY is calculated in a range of 1.91 and 2.49 eV. All these performed studies prove that the optical properties of GDY significantly differ from graphene owing to different band structures.

### 2.5. Black Phosphorus

Black phosphorous (BP) materials have attracted considerable interest in nanoelectronic devices due to their intrinsic semiconductor behavior; however, their poor availability and solution processability are the main complications for producing large amounts of high-quality 2D BP, restricting its broader application in nano- or micro-electronics [78]. BP is a layered thermodynamically stable form of phosphorous (Figure 6a) [79,80]. Each atomic layer of BP interacts together with van der Waals forces in the place of covalent or ionic bonds. Bulk BP can be exfoliated into 2D mono- or few-layer sheets by applying driving forces to break the weak van der Waals interactions (Figure 6b) [79]. BP is an ambipolar semiconductor with a direct bandgap of 0.39 for its bulk, and a tunable direct bandgap for the 2D layered state depending on the interlayer stacking pattern and the number of layers. The number of layers plays an important role in the electronic, optical, and thermal properties of BP since, for example, the monolayer BP has a direct bandgap of 2 eV, decreasing with increasing the number of layers (Figure 6c) [78,81]. As illustrated in Figure 6d, the direct bandgap character of the 2D layered BP systems retains for the thicknesses up to 5L, unlike MoS_2_ (for example) with a direct bandgap character maintaining only for its monolayer [82]. In addition, unlike other 2D materials, 2D BP possesses a strong in-plane anisotropy providing a high carrier mobility (about 1000 cm^2^/Vs) and thermoelectric performance. The research on 2D BP materials is still in early stages; however, they have already been introduced as a promising candidate for transistors, supercapacitors, batteries, and solar cell applications [78,81,83,84].

Unique the properties of thickness-dependent bandgap, the high carrier mobility and ambipolar conduction features of 2D BP materials make them a promising material among the growing family of 2D materials for developing high-efficiency, low-cost solar cells. The power conversion efficiency of 20% was predicted for the excitonic and heterojunction solar cells based on 2D BP materials [85,86]. The application of 2D BP nanosheets as both the hole-transporting layer (HTL) and the electron-transporting layer (ETL) of organic photovoltaic (OPV) devices has been reported [87]. 2D BP derivatives can be employed as the interface of heterojunction solar cells (HJSCs) with the power conversion efficiency of 8.3% and the hole-transporting layer (HTL) of perovskite solar cells (PSCs) with a power conversion efficiency of 16.69% [88,89].

### 2.6. Other Materials

There is a growing interest in both the synthesis of 2D nanomaterials other than graphene and the exploration of their potential properties and applications. In this context, layered materials represent a considerable source with regard to 2D systems. Recently, a feasible procedure to produce novel 2D materials has been developed based on a combination of chemical modification and the sonication technique [2,90]. Using appropriate acid solutions, the transition-metal carbides, belonging to the MAX phase family have been exfoliated into 2D transition metal carbides and nitrides (the so-called MXenes), wherein the MAX phase family constitutes layered solids with the chemical formula M*_n_*_+1_AX*_n_* (*n* = 1–3) with M representing an early transition metal (e.g., Sc, Ti, V, Cr, Zr, Nb, Mo, Hf, and Ta), A denoting A-element atoms (i.e., mostly group 13 and 14 elements of the periodic table), and X to be carbon and/or nitrogen [2,91,92]. 

After acid treatment, the MAX phases are exfoliated into 2D M_2_XT*_x_*, M_3_X_2_T*_x_*, or M_4_X_3_T*_x_* MXenes (where T*_x_* represents the surface terminations, e.g., OH, O, F, or Cl). These 2D systems have been named as MXenes for two principal reasons: they originate from the MAX phases following the removal of A-element atoms and they are structurally analogous to graphene [92,93,94,95]. Figure 7a demonstrates a typical structure of the MAX phase and their derived 2D MXene family (surface terminations are not shown here). The exfoliation of MAX phases is possible because, in some cases, the chemical bonds between the M-A elements are weaker than the M-X bonds; therefore, the selective etching of A-element layers from the bulk MAX is possible through acid treatment. 

As shown in Figure 7b, in a 2D flake of MXene, *n* layers of element X (elements in gray) are sandwiched between *n* + 1 layers of element M (elements in blue), with the potential surface termination elements that are bonded to the outer M layers shown in orange. The first MXene (Ti_3_C_2_T*_x_*) was produced in 2011 [94]. Subsequently, more than 30 different MXene compositions have been synthesized (marked in blue in Figure 8), while many more have theoretically been predicted based on computational methods [95,96,97] (marked in gray in Figure 8).

The MXene family has been expanded by the mixing of two transition metals in an MXene structure in two different forms, which has provided an attractive opportunity for fine-tuning the properties of the composition by varying the contents of the M and/or X elements. First, a random arrangement of two different transition metals can be formed in the M-layers (solid solution), e.g., (Ti,V)_2_CT*_x_* (marked in green in Figure 8). Second, transition metals can form ordered structures in a single 2D MXene flake as marked in red in Figure 8, either by creating single or double layers of one transition metal sandwiched between the layers of a second transition metal (for *n* ≥ 2), e.g., Mo_2_TiC_2_T*_x_*, or as in-plane (*n* = 1) ordered structures, e.g., (Mo_2/3_Y_1/3_)_2_CT*_x_*. The first ordered MXenes were reported in 2015 [98] and many other compositions have since been produced. Considering many compositional varieties of MAX phase compounds, it is expected that a large number of 2D MXenes with unprecedented properties will be produced in the near future. The first application of MXenes was in energy storage, and this remains the field of greatest MXene-related activity. The latest exciting development is in the application of MXenes in biosensors, photothermal therapy of cancers, dialysis, neural electrodes, and theranostics [99,100]. Many applications of MXenes are based on their exceptional combinations of properties, including their high mechanical properties and electrical conductivity [93,101,102], hydrophilic features attributable to their functionalized surfaces providing the opportunity to bond with different species [100,103], absorptions of electromagnetic waves [104], and assistance in creating stable colloidal solutions in water [105]. Another area in which MXenes excel in comparison with other nanomaterials is in electromagnetic applications such as printable antennas and electromagnetic interference shielding [106]. In certain fields, including electronic and structural applications, the use of MXenes has been explored theoretically but with the comparatively limited experimental investigations, and predicted properties such as topological insulation/ferromagnetism have not yet been examined experimentally.

This novel family of 2D MXenes, which is breaking new ground in the field of materials research, promises new scientific and technological horizons because 2D transition-metal-based systems represent an excellent field for the exploration and exploitation of the internal degrees of freedom of electrons—charge, orbital, and spin—and their interplay in relation to fundamental research and device applications. Research on 2D MXenes is progressing swiftly, with more than 3000 papers published on this topic since 2011 to date, half of which were published in 2019 and 2020 according to Web of Science, evidencing a rapid expansion of the MXene field. The breakthrough in the synthesis of 2D MXenes from layered MAX phases has raised the possibility of the synthesis of novel 2D materials from other layered bulk materials. More recently, it has been revealed that Al-containing transition-metal borides, namely MBene with MAB phase, can also be exfoliated into 2D structures [107]. A large family of 2D MXenes possesses various electronic properties duo to the different possible combinations of M, A, and X elements. Generally, all pristine MXenes are metallic; however, some semiconducting display behavior upon surface functionalization. The properties of MXenes are discussed in Section 4.

## 3. Fabrication of Organic Photovoltaic and Silicon Solar Cells

With the rapid development of the semiconductor industry and increasing demands for renewable energy sources, OPV and silicon solar cells have drawn considerable attention. Compared to that of traditional materials based solar cells, these two mentioned solar cells based on 2D materials, in particular, MXenes based solar cells, have attracted considerable attention due to many advantages, such as excellent thermoelectric property, high quality and large scale synthesis methods, high carrier mobility at room temperature, large electrical conductivity, and so forth. Meanwhile, based on these mentioned superior properties, solar cells, whose channel or functional materials consist of 2D materials, provide much better photovoltaic parameters than that of traditional materials as well, in particular, power conversion efficiency, reported as high as 26.5%, is nearly three times greater than that of single-walled carbon nanotube-based silicon solar cells. In this regard, we summarize and highlight some recent representative investigations regarding the fabrication processes of these two mentioned solar cells based on 2D materials.

### 3.1. Organic Photovoltaic Solar Cells

As one of the 3rd generation solar cells, organic solar cells have attracted considerable interest since they are low-cost, environmentally friendly, versatile, suitable materials, with higher PCE than traditional silicon-based solar cells, as well as their compatibility with printable and flexible substrates. In terms of organic solar cells, it can be mainly divided into four categories: bulk heterojunction (BHJ), modern vertical alternating multilayer, nanostructure hybrid, and wearable/printable organic solar cells. In this section, we will highlight some recent representative investigations regarding the fabrication of the abovementioned organic solar cells.

BHJ organic solar cells commonly consist of an anode, a photo-active layer, an organic polymer, a hole transport layer (HTL), an electron transport layer (ETL), and a cathode. Organic solar cells possess a sandwich structure and build vertically on indium tin oxide (ITO) glass substrates (as shown in Figure 9a) [108]. To fabricate a BHJ organic solar cell, cleaning and deposition are two key procedures; further, organic matter or dust particles must be removed from the employed transparent glass-ITO substrates using a series of solvents and detergents to guarantee efficient solar absorption. Ultrasonic and ultraviolet–ozone treatments are also applied to further removal of purities. Additionally, under ultraviolet–ozone treatment, the hydrophilicity of the glass-ITO surface can be improved as well, being mainly due to the increased oxygen concentration. Then, HTL is deposited between the active layer and anode. Noticeably, to provide more channels for the holes that exist at the interface of HTL and the active layer, the surface of HTL should be smooth. After spin-coating and vacuum thermal evaporation treatments, the electron transport layer and cathode are deposited. Based on these fabrication processes, Che et al. fabricated a high-performance organic tandem solar cell that involves both non-fullerene acceptors and fullerene sub-cells, as shown in Figure 9b [109]. The glass-ITO substrates are chemically cleaned using a set of solvents and detergents, followed by an ultraviolet–ozone treatment that lasts for 10 min. The blended active region and active layers for the front cell are grown through vacuum thermal evaporation; meanwhile, a vacuum chamber connected to gloveboxes is employed to grown solution-processed layers. These materials are then deposited in sequence onto the surface of organic materials. Under the spin coating process, the transparent organic materials, poly(3,4-ethylenedioxythiophene): poly(styrenesulfon ate) and non-fullerene active layer, are coated onto the glass-ITO substrates at 5000 and 2000 revolutions per minute (rpm) for 60–90 s, respectively. The non-fullerene material is dissolved first in chlorobenzene:chloroform, and the mixture is stirred overnight on a hot plate at 65 °C. In the deposition process, the samples are transferred to the vacuum chamber for the deposition of the Ag cathode and TmPyPB. An antireflection coating is grown onto the transparent substrates to enhance the absorption efficiency further. To realize a low refractive index based on the beam direction, the SiO_2_ and MgF_2_ are usually grown and deposited on the substrate with an angle of 85°. The effective absorption wavelength of the device is extended to 950 nm, and the PCE is as high as 15.0 ± 0.3%. Flexible organic solar cells have drawn considerable attention since their adoption in various applications, such as performing arts, military clothing, fitness, and sports. Recently, Li et al. proposed an entirely spray-coating-based method to fabricate textile organic solar cells, as shown in Figure 9c [110]. The bottom Ag-electrode spray-coated layer was heated with a temperature of 150 °C for 5 min to achieve a 100 nm thin silver bottom electrode. Subsequently, the photovoltaic active layer was placed on the surface of the ZnO-NP layer to obtain dry flakes; the substrate with photovoltaic active layer was then heated at 90 °C for 10 min and then the PEDOT: PSS layer was spray coated on the photovoltaic active layer and baked with a temperature of 100 °C after a cooling process. Finally, the top Ag-nanowire electrode was deposited on the surface of the PEDOT: PSS layer. After the baking process at 100 °C, the semitransparent top electrode could be realized. To avoid short circuits, the size of the top Ag-nanowire electrode should be small. The screen-printed silver paste was employed to contact the flexible polyester cotton woven textile bottom substrate. To achieve a 1mm thick encapsulation layer, all mentioned layers were encapsulated through a doctor blading process. Beyond commonly used glass-ITO substrates, the paper substrate is thought to be another promising substrate due to being low cost, ecologically friendly, and having the capacity for large-scale and high-quality synthesis processes. Motived by these advantages, Rawat et al. demonstrated environmentally friendly organic solar cells on paper substrates [111]. The polyvinyl formal was employed as a smoothening layer and was knife coated onto a glossy paper substrate (Figure 9d). Through the vacuum thermal evaporation process, an Ag-electrode was deposited on the polyvinyl formal surface. Meanwhile, the ZnO layer was formed by a sol–gel process, under spin-coated (3500 rpm for 1 min) and anneal (150 °C for 15 min) processes, a 30 nm thick ZnO layer was achieved. Then, the fabricated device was transferred into a nitrogen-filled glove box for depositing a photoactive layer. Via spin coating (1000 rpm for 30 s) and annealing (110 °C for 5 min) processes, a 90 nm thick photoactive layer was achieved. The top transparent electrode was formed through spin coating PEDOT:PSS and annealed (100 °C for 10 min) processes. By employing an epoxy with the tapes of a Tera-barrier, the whole device was encapsulated, wherein the measured PCE of the device was 6.44 %, which was the highest value among all other organic solar cells on paper substrates. Long-term stability is another key performance parameter of solar cells; to improve this determined parameter, abundant efforts have been performed. Organic and inorganic hybrid structures are an effective means due to their cost-effectiveness and extremely simple fabrication process. Subramani et al. presented a novel long-term stability organic and inorganic hybrid solar cells [112]. The fabrication process is shown in Figure 9e, wherein a silicon wafer was employed to fabricate Si nanostructures through the etching (immerse in H_2_O_2_ and HF for 25 s) process. Then, HNO_3_ and HF solutions were used to remove the silver and Si oxide, respectively. To efficiently decrease metal contamination and surface defects, chemical polishing etching was carried out. 

Meanwhile, through this process, silicon nano-holes were modified into tapered silicon nano-tips. After this, the silicon nano-tips were placed on a 50 °C hot plate; via drop-coating and spin-coating means, a TED-Li/Si-nanotip hybrid layer was deposited onto the silicon nano-tip samples. Ag (250 nm)/Ti (50 nm) back electrodes and finger-shaped Ag film-top electrodes were evaporated and deposited on the rear and front sides of the device, respectively. The fabricated device exhibited excellent long-term stability with a 10 % loss in one month, which is beneficial for practical applications.

### 3.2. Silicon Solar Cells

Nowadays, silicon solar cells are the most explored and established new energy material due to several aspects: (i) silicon is abundant with efficiently suppressed manufacturing cost, (ii) quite advanced and mature fabrication processes, (iii) the extraordinary environmental stability and wide operating temperature range enabling silicon solar cells to dominate the solar cell market, (iv), and a significantly improved power conversion efficiency (PCE) through hybridization with other solar materials. Meanwhile, to pursue high PCE and high-performance silicon solar cells, attentions have been drawn to develop novel silicon solar cells; thus, in this section, we present some recent representative investigations regarding the fabrication of novel silicon solar cells.

To efficiently suppress the cost and develop various flexible, functional silicon solar cells, silicon with a thickness of 100 μm or less is usually proposed; however, based on conventional fabrication means, in particular, high-temperature soldering process, undesirable interfacial stress is introduced between silicon solar cells and metal ribbons having negative and positive contributions to the PCE and breakage loss, respectively. To address the challenges mentioned above, Shin et al. proposed a novel layup-only modulization (LOM) method to achieve high-performance silicon solar cells with 100 μm silicon solar module [113] wherein the LOM fabrication method can be divided into six steps, as shown in Figure 10a. After a purification process, a set of metal ribbons were firstly coated on the front side of ethylene-vinyl acetate (EVA) substrate through a low melting temperature soldering treatment (139 ℃). At the rear side, a silicon solar cell with a thickness of 100 μm was laid, and then a rear EVA flake with metal ribbons was pre-attached, and a back sheet was laid in sequence. After this fabrication process, the fabricated thin silicon solar cell was placed into a vacuum laminator with a temperature of 160 ℃. Noticeably, by using the LOM fabrication method, the power loss of 100 μm silicon solar cells can be maintained with a small value of −0.8 ± 1.6%. Another route to achieve high PCE solar cells is employing silicon heterojunction methods providing extraordinary surface passivation since the hydrogenated intrinsic amorphous silicon films. Basnet et al. [114] presented a high PCE solar cell based on silicon heterojunction wherein an upgraded metallurgical-grade silicon wafer was placed into a surfactant additive (GP Solar, ALKA-Tex) and potassium hydroxide (KOH) solution. After a chemical cleaning process, the buffered oxide etching was applied to achieve the hydrophobic behavior of the wafer surface. To realize electron and hole contact, 8 nm of intrinsic a-Si:H(i) was deposited on the front and rear sides, respectively. Meanwhile, 4 nm of phosphorus-doped a-Si:H(*n*) using phosphine and 11 nm of boron-doped a:Si:H(p) using trimethylboron were employed as doping gases, as shown in Figure 10b. After the a-Si:H(i) layer was deposited on both rear and front sides, an in situ hydrogen plasma treatment was applied for 15 s to further improve the a-Si:H(i)/c-Si interface passivation. The efficiency was measured to be 3.2% higher than that of solar cells without a silicon heterojunction wafer and with a value as high as 21.2%. Additionally, Qian et al. demonstrated p-doping single-walled carbon nanotubes-based silicon solar cells with the highest stable efficiency than that of other presented carbon nanotube-based silicon solar cells (Figure 10c) [115]. The n-type Si was placed into the H_2_O:NH_4_OH:H_2_O_2_ = 5:1:1 mixture solutions and heated to 70 °C for half an hour to remove organics and dust from the surface. Then, a 5 M NaOH solution was employed to remove SiO_2_ layers on both sides of the n-type Si substrate by the heating process. After this, combined with distilled water rinsing treatment, the etched n-type Si substrate was chemically cleaned by H_2_O:NHCl:H_2_O_2_ = 5:1:1 mixture solutions. Under high-purity nitrogen gas blow disposition for 3 s, the front Ti (10 nm)/Pt (55 nm) and back Ti (10 nm)/Pt (55 nm) electrodes were sputtered onto the front and back sides of the n-type Si substrate, respectively. The single-walled carbon nanotubes were coated on the n-type Si surface and surrounding electrodes through a dry-transferring method. To further enhance the contact between transferred flake and substrate, ethanol was dropped onto the fabricated device. The measured PCE of the device was up to 14.4% and, with unprecedented environmental stability last for 120 days even under severe conditions. Recently, integrating polycrystalline or amorphous semiconductors, such as perovskites and cadmium telluride with silicon has attracted considerable attention since the absorption wavelength range can be significantly extended without considering lattice mismatch during the deposition process, and higher tolerance on defects, which is discussed in detail in Section 5.4.

## 4. Synthesis and Property of 2D MXenes

The synthesis, properties, and applications of novel 2D materials have become one of the most exciting areas of interest in science and technology. Single layers of graphene, boron nitride (BN), transition-metal chalcogenides (MoS_2_, WS_2_, etc.), and phosphorene have been successfully obtained from their bulk van der Waals layered structures. Recently, it has been shown that by using a combination of chemical exfoliation and sonication, the synthesis and mass production of 2D materials from three-dimensional (3D) layered compounds with chemical bonding between the layers is also feasible. In this regard, many members of the MAX phase family (more than 130 reported to date) can be exfoliated into 2D MXenes by selective etching of the A element layers [96,116,117,118,119,120]. 

The process of making MXenes and the separation of M_n+1_X_n_ layers by mechanical shearing of MAX phases is impossible due to the metallic nature of the M–A bonds [120,121,122,123]. However, since the chemical bonds between M–A elements are more chemically active than the stronger M–X bonds, it provides a condition to selectively etch off the A-element layers of MAX phases to synthesize MXenes [124,125,126]. In this regard, it has been demonstrated that using different synthesis techniques (e.g., hydrofluoric acid solutions), some members of the MAX phase family can be exfoliated into 2D MXenes [127,128,129]. During the exfoliation process, the A element layers are removed from MAX phases resulting in the formation of various 2D M_n_X_n+1_, M_2_M’X_2_, M_2_M’_2_X_3_, or (M_2/3_M’_1/3_)_2_C MXenes, shown in **Figure 11**. Depending on the type of chemical environment, a mixture of F, O, OH, or Cl groups terminate the surface of MXenes.

Different synthesis techniques have been applied to produce 2D MXenes. These techniques are listed here and are explained in detail in the following sections. Top-down synthesis methods have been primarily employed as the main route to produce 2D MXenes, which refers to the synthesis techniques where 2D MXenes will be delivered from their precursors. The most common top-down technique is to etch the A-element layers with aqueous hydrofluoric acid (HF). Another commonly used technique to produce MXenes is to use a mixture of lithium fluoride (LiF) and hydrochloric acid (HCl) to make the in situ formation of HF through a reaction between LiF and HCl. Other techniques are also reported to etch the A-element layers, such as mixing HF with other acids (e.g., HNO_3_), including fluorine-containing organic solvents, and providing a reaction between the MAX phase and molten salts (e.g., ZnCl_2_), where the A-element layers will be replaced by the Zn and then removed using HCl. Further, MXenes have been produced from non-MAX phase precursors. In addition to the top-down methods, MXenes are produced from the bottom-up synthesis methods, such as chemical vapor deposition (CVD).

It should be noted that the synthesized MXenes are contained intrinsic defects [2,103,130]. The quality, defects, and surface functional groups in MXenes depend on the etching and delamination conditions. Generally, to produce MXene flakes with higher quality and lower defect concentration, milder etching conditions should be applied [131,132,133]; however, the etching conditions for the successful synthetization of different MXenes are dependent on the containing elements and the electronegativity between them. For example, V-, Nb-, Ta-, and Mo-based MXenes require more severe etching conditions [131]. In addition, the concentration of each type of surface functional group depends on the etching conditions in addition to the M element and storage of the MXenes produced [2]. The MXenes synthesized with lower concentrations of HF etchants are covered with a more significant percentage of oxygen functional groups.

### 4.1. Synthesis Methods

#### 4.1.1. Top-Down Synthesis Approaches

2D MXenes are usually produced from top-down methods by selective etching and exfoliation of their MAX precursor phases possessing strong M–X and weak M–A bonding. Unlike graphene produced by mechanical exfoliation of graphite, 2D MXenes cannot be produced by mechanical exfoliation due to the metallic nature of M–A bonding; however, M–A bonds are more chemically active than the more robust M–X bonds, which provide exfoliation of MAX phases by selective etching of A-element layers [2,93,96].

Since the production of the first MXene (i.e., titanium carbide (Ti_3_C_2_)) by selective etching of Al layers from its precursor (i.e., Ti_3_AlC_2_ (MAX)) using hydrofluoric acid (HF) [94], top-down synthesis routes using HF, or in situ formation of HF using fluoride solutions have been widely applied to synthesize many MXenes from their layered MAX phases. Although fluoride-based etchants have mainly been used for top-down synthesis methods due to the selectivity to etch the A element, some other alternative top-down synthesis methods have also been applied using molten salts, alkaline solutions, and hydrothermal treatments. It is also plausible to synthesize some MXenes from non-MAX phase precursors; however, top-down selective etching is the fundamental method of MXene production. A schematic of top-down selective etching of A layers from MAX and non-MAX phases to synthesize M_2_XT*x*, M_3_X_2_T*x*, and M_4_X_3_T*x* MXenes is illustrated in Figure 12. It is worth noting that the concentration of the surface terminations (–OH, –O, –F, and –Cl terminations), which plays an essential role in the properties of MXenes (e.g., electronic band structure, optical absorption, and chemical reactivity), depends on the synthesis method.

Selective etching with HF has been employed for the production of several MXenes that leads to weakening the M–A metallic bonds while the strength of the M–X bonds is retained, following to etching of the A layer (typically A = Al) from the MAX phases (M = Ti, Nb, Ta, V). HF is a highly selective etchant with a potential to selectively remove the A layer in a temperature ranging from room temperature up to 55 °C by adjusting the HF concentration and etching time; however, the etching conditions differ for different MAX phases, depending on the M, A, and X elements, MAX structure, and bonding strength between the elements [135,136]. Generally, as the M–A bonds are more robust, a longer time and stronger etching are required. In this regard, for the Al-containing MAX phases, metallic M–Al bonds are more robust for the M elements with a more significant number of valence electrons, following stronger etching; therefore, the etching conditions for every MXene are different, affecting the quality of MXene, such as defect concentration and surface functionalization. More information regarding etching conditions for different MXenes produced can be found in Reference [2].

HF is not an environmentally friendly etchant, and as an alternative for HF, a mixture of a strong acid (e.g., HCl) and a fluoride salt (e.g., LiF) has been used to make MXenes [131,137]. Indeed, in situ HF etchants can be formed via the reaction of HCl and LiF for selective etches of the A atomic layers of MAX phase precursors. Another route to create in situ HF etchants is through the hydrolysis of bifluoride salts (e.g., NaHF_2_, KHF_2_, and NH_4_HF_2_) [130,138]. For example, Ti_3_C_2_T*_x_* and Mo_2_CT*_x_* have been synthesized using in situ HF etchants formed via the reaction of LiF and HCl [131,137]. In addition, NaHF_2_, KHF_2_, and NH_4_HF_2_ have been employed to form in situ HF etchants in the synthesis of Ti_3_C_2_T*_x_* for the selective etching of Al layers from Ti_3_AlC_2_ [130,138].

Selective etching with in situ HF-forming etchants provides larger interlayer spacing and thus weaker interaction between MXene layers compared to employing pure HF etchants, attributing to the intercalation of cations (e.g., Li^+^) or bifluoride salts used to make the etchants. This can be introduced as the advantage of in situ HF forming over pure HF etching leading to the delamination of the MXene without further steps to attain single- or few-layer flakes [2].

Selective etching of A layers from nitride-based MAX phases with pure HF or in situ HF-forming etchants is not successfully plausible. This is mainly because the difference between the M–X and M–A bonding strength in nitride MAX phases is lower than that of carbide MAX phases, following the higher formation enthalpies of nitride MXenes from nitride-based MAX phase than that of carbide MXenes from the carbide-based MAX phase [96]. Molten salts (e.g., KF, LiF, and NaF), as an alternative to HF etchants, have been used for the selective etching of A layers of nitride-based MAX phases. For example, Ti_4_N_3_T*_x_* has been synthesized through selective etching of Al layers from Ti_4_AlN_3_ using molten salts [139]. To overcome the difference between the M–X and M–A bonding strength in the MAX phase, more energy is required for the reaction; therefore, the molten salt and Ti_4_AlC_3_ mixture was heated to 550 °C in argon for 30 min [139]. Recently, it has been shown that the selective etching of Al layers from the Ti_3_AlC_2_ MAX phase for the synthesis of Ti_3_C_2_T*_x_* MXene using a salt-based method has been improved with the help of surface acoustic waves (SAWs), which produce protons. Then, the combination of these protons with fluorine ions from LiF helps to selectively etch the MAX phase into MXene [140].

Since alkali has the potential to make strong bonding with Al, alkaline etchants could be proficient in etching layered MAX phases, which hydrothermal treatment (i.e., higher temperatures and/or pressures) are required to eliminate Al in an alkaline environment. For example, Ti_3_C_2_T*_x_* has been synthesized without fluorine surface termination using the alkali-assisted hydrothermal treatment of Ti_3_AlC_2_ MAX phase in concentrated NaOH [141]. In this synthetization process, known as the Bayer process, Al layers in the Ti_3_AlC_2_ MAX phase have been turned into aluminum oxide/hydroxides using concentrated NaOH solutions under high temperatures and pressures. Then, the aluminum compounds have been dissolved into the alkali solution and making Ti_3_C_2_T*_x_* MXene terminated with only OH and O surface functions [141].

HF-based etchants are efficient to weaken the M–A bonds with A = Al more than other A elements (e.g., Si, Ga, and Ge), following the exfoliation of Al-containing MAX phases, mainly owing to dissolution and etching kinetics by HF differ between various A layers. That is why the Ti_3_AlC_2_ MAX phase has been employed to synthesize Ti_3_C_2_T*_x_* MXene, instead of, for example, Ti_3_SiC_2_; therefore, non-MAX-phases, including A layer other than Al, have been used to produce MXenes. For instance, Mo_2_CT*_x_* is introduced as the first MXene synthesized by etching Ga layers from Mo_2_Ga_2_C [131]. Another example is Zr_3_C_2_T*_x_* made from Zr_3_Al_3_C_5_, as a non-MAX-phase precursor, by selective etching Al_3_C_3_ layers [142]. This is mainly because etching the Al–C units is energetically more promising than only the Al layers in Zr_3_Al_3_C_5_; however, Ti_3_C_2_T*_x_* MXene has been recently synthesized from a Si-containing MAX phase using a mixture of HF and oxidant as etchants for selectively etching of a non-Al layer (i.e., Si layer) from Ti_3_SiC_2_ [143].

#### 4.1.2. Bottom-Up Synthesis Approaches

In addition to the top-down synthetization methods, MXenes have been formed using bottom-up methods, such as chemical vapor deposition (CVD) [144]. Bottom-up strategies can make bare/pristine 2D MXenes. For example, a thin layer (2–3 nm thick) of 2D α-Mo_2_C with a lateral size of 100 µm has been produced using the CVD technique from methane on a molybdenum foil with a bilayer substrate of copper foil [145]. In addition to Mo transition metal, other transition metals of W and Ta have been formed into ultrathin WC and TaC crystals using the CVD technique. Unlike many efforts made on top-down routes, bottom-up strategies are rare. The main advantage of this method is that the MXenes produced possess few defects and large lateral size, providing a proper condition to investigate their intrinsic properties; however, those bottom-up techniques should be further explored for the synthesis of MXene monolayers [144,145].

### 4.2. Stability

It has been revealed that monolayer MXene flakes cannot be stable in environments containing oxygen and water or being at high temperatures, while they can be stable in dry air or oxygen-free degassed water; therefore, it is required to store MXenes cooled in an environment away from oxygen and water, e.g., an oxygen-free dark environment. To illustrate the point, Ti_3_C_2_T*_x_* flakes can be fully oxidized in several days if the solution in water is exposed to air. Generally, the oxidation effects on the MXene flakes appear on the edges at the beginning, where metal oxide nanocrystal (e.g., TiO_2_ in the case of Ti_3_C_2_T*_x_*) forms, then the oxidation affects growth on the whole surface [146,147]. It is worth noting that the manufacturing procedure will affect the environmental stability so that the single flakes produced with higher quality retain higher environmental stability [132]. Although the stability of MXenes at high temperatures has not been well recognized, the effect of MXene composition and the atmosphere seems to be dominant at elaborated temperatures. For example, the stability of Ti_3_C_2_T*_x_* in an Ar atmosphere at 500 °C has been reported; however, it has been revealed that some TiO_2_ crystals form on the edges of the flakes [146]. Additionally, the Ti_3_C_2_ structure has been well kept in an Ar atmosphere even at 1200 °C, but MXene phase transformation to cubic TiC*_x_*, as the most stable phase in the nonstoichiometric TiC phase diagram at 1200 °C, was detected in XRD observations [146]. The following example is Ti_2_CT*_x_,* which has a stability under various inert atmospheres at 250 °C, confirmed by [147]; however, Ti_2_CO_2_ has been proved to be stable in Ar/H_2_ at 1100 °C, which is above the phase stability limit of nonstoichiometric Ti_2_C [148]. Further, Zr_3_C_2_T*_x_*, as another example, has a confirmed thermal stability. Unlike Ti_3_C_2_T*_x_* transforming to cubic carbide, Zr_3_C_2_T*_x_* maintains its 2D structure under vacuum conditions at higher temperatures of 1000 °C. Since the structure of Zr_3_C_2_T*_x_* is more energetically favorable than bulk ZrC, it is more stable at high temperatures than Ti_3_C_2_T*_x_* confirmed metastable relative to bulk TiC [105]. The higher thermal stability of Zr_3_C_2_T*_x_* introduces it as a promising candidate for high-temperature applications.

### 4.3. Properties

The combination of a wide variety of transition metals (i.e., M) and A elements with carbon and nitrogen in MXenes provides unique targeted properties, introducing MXene as promising versatile materials for a wide variety of applications (see Figure 13). Although many efforts have been made to realize the properties, still, a lot of investigations are yet to be conducted to fully understand their properties.

Investigations on the mechanical properties of MXenes have introduced Ti_3_C_2_T*x* MXene as the strongest solution-processed 2D materials beyond TMDs and graphene oxide and the third-strongest 2D nano-sheet after bare graphene and hexagonal boron nitride [4]. The Young’s modulus of 330 GPa for a monolayer Ti_3_C_2_T*_x_* MXene was measured using atomic force microscopy nano-indentation. Although some other studies have explored the mechanical properties of MXenes, further studies are still demanded [4,93,149]. All pristine MXenes have a metallic behavior similar to their MAX precursors; however, those surface terminations affect their electronic properties as functionalized MXenes have been predicted to be either metals or semiconductors, depending on the containing elements and surface functional groups [150]. The electrical properties of MXenes are discussed in more detail below.

It has also been confirmed that MXenes possess excellent optical properties [2]. For example, a 1 nm thick Ti_3_C_2_T*_x_* MXene can absorb 3% of visible light. By changing the elements in MXenes, the optical properties of this family of materials can be tuned, as discussed below.

#### 4.3.1. Electrical Properties

A large family of 2D MXenes possesses various electronic properties owing to different possible combinations of M, A, and X elements. Generally, all pristine MXenes are metallic; however, some display semiconducting behavior upon surface functionalization. Theoretically, the electronic properties of 2D MXenes are divided into two groups of (a) topologically trivial and (b) topologically nontrivial metal/semimetal or semiconductor, in which the strength of spin-orbit coupling (SOC) play a crucial role [93,101,150,151].

Topologically Trivial Metals and Semiconductors. The metallic behavior is theoretically predicted for most of the functionalized MXenes; however, several MXenes are predicted to be semiconducting. For example, Sc_2_CT_2_ (T = F, OH, and O), Ti_2_CO_2_, Zr_2_CO_2_, and Hf_2_CO_2_ display the semiconducting behavior. The bandgap of these MXenes is predicted in a range of 0.45 and 1.8 eV for Sc_2_CT_2_, depending on the surface functionalization; 0.24 eV for Ti_2_CO_2_; 0.88eV for Zr_2_CO_2_; 1.0 eV for Hf_2_CO_2_ (Figure 14a–f). All of these semiconducting MXenes possess an indirect bandgap and only Sc_2_C(OH)_2_ possesses a direct bandgap. Generally, the MXenes with the M elements of the same group in the periodic table (the elements with the same number of electron valance) reveal similar metallic to semiconducting behavior upon the same surface functionalization. For example, Ti-, Zr-, and Hf-based MXenes with the same type of surface function in which the M elements are in the same group of the periodic table display the same electronic behavior. Further, F and OH surface functions have the same effect on the electronic structure of MXenes since they will need only one electron to fill their outermost atomic orbital shell. In this regard, the O group will affect the electronic properties of MXenes differently since oxygen needs two electrons to fill its outermost atomic orbital shell, then two electrons will be transferred from the surface to the O group [101,150]. As an example, the projected density of states and band structures of pristine Ti_2_C, Ti_2_CF_2_, and Ti_2_CO_2_ are shown in Figure 14g–i, in which Ti_2_C MXenes becomes semiconducting when functionalized with oxygen. It has also been investigated that applying strain and external electric field have a considerable effect on the bandgap of Ti_2_CO_2_ and Sc_2_CO_2_ [152,153]. Further, the electronic properties of in-plane ordered double transition metals MXenes reveal that (Mo_2_*_/_*_3_Y_1_*_/_*_3_)_2_C and (Mo_2_*_/_*_3_Sc_1_*_/_*_3_)_2_C MXenes have a semiconducting behavior upon the oxygen surface functionalization with indirect bandgaps of 0.45 and 0.04 eV, respectively [154].

Topologically Nontrivial Semimetals and Semiconductors. Because of containing 4*d* and 5*d* transition metals in many MXenes, the spin-orbit coupling (SOC) effect could play a crucial role in their electronic structures. For example, without considering the SOC, M_2_CO_2_ (M = Mo, W), M_2_M’C_2_O_2_ (M = Mo, W; M’ = Ti, Zr, Hf), and Ti_3_N_2_F_2_ display semiconductor-like behavior with a zero-energy gap (semi-metallic), as shown in the upper panels in Figure 15, where the top-most valence band and the lowest conduction band meet at the Γ point. As can be seen from the projected band structures, the *d* orbitals of the transition metals have a substantial contribution near the Fermi energy. However, considering the SOC, the top-most valence band and the lowest conduction band at the Γ point are separated, and the bandgap is opened at that point (lower panels in Figure 13), displaying semiconductor-like behavior with indirect bandgaps. As can be seen, the SOC has a varied effect on the bandgap, and the bandgap can be opened more significantly as the SOC is larger. It is reported that the bandgap of these MXenes can be opened as 0.194 eV (0.472 eV) for W_2_CO_2_ and 0.285 (0.401 eV) for W_2_HfC_2_O_2_ within the GGA (hybrid functional HSE06) while the bandgap of Ti_3_N_2_F_2_ is approximately 0.02 eV.

#### 4.3.2. Optical Properties

As stated earlier, because of the unique properties of 2D transitional metal dichalcogenides (TMDs; e.g., MoS_2_, WS_2_, WSe_2_, MoTe_2_), graphene, graphdiyne (GDY), and black phosphorus (BP), they have been introduced to the optics and electronics community. In addition to this category of 2D materials, MXenes have attracted a significant amount of attention from the community because of their tunable properties and variety of applications. Many MXene compositions have been predicted and synthesized, introducing this family of 2D materials as the fastest developing and the most diverse 2D materials [96,100,122,123,124]; however, among these compositions, the optical properties of Ti-based MXenes have mostly been investigated [155,156]. The optical response (complex dielectric function (*ε*), reflection, absorption, and electron energy loss function) of the 2D Ti_n+1_X_n_ (X = C, N and *n* = 1, 2) MXenes are, for example, shown in Figure 16. Monolayer Ti_n+1_X_n_ (X = C, N and *n* = 1, 2) MXenes possess a hexagonal space group symmetry including three and five atomic layers for Ti_2_X and Ti_3_X_2_, respectively; therefore, because of the symmetry, the complex dielectric function (*ε*) tensors would turn into only three nonzero components (ε_xx_(ω) = *ε*_yy_(ω) and *ε*_zz_(ω)) upon the excitation of an in-plane (*E*||*z*) and out-of-plane (*E*||*x*) electric field (Figure 16a). There are direct contributions between the peaks in the Im *ε* and the different inter- and intraband electronic transitions. The real part of the *ε,* Re *ε*, is calculated using the *Kramers–Kronig* relations [100]. Knowing the frequency-dependent complex dielectric function, other optical parameters such as the reflection and absorption are calculated. The absorption spectrum (*A*) is proportional to the total contribution of interband transitions from filled valence band states to empty conduction band levels (Figure 16b).

In addition, the effect of surface functionalizations on the optical properties of Ti_3_C_2_T_2_ MXene was explored [156]. The theoretical predictions of the real and imaginary parts of the frequency-dependent dielectric functions were investigated, the type of surface functions has a considerable effect on the dielectric function; in addition, absorption spectra for small and larger ranges of photon energy have been calculated for Ti_3_C_2_T_2_ MXene. The results indicate considerable differences between the absorption spectra upon different surface functions, mainly in the lower photon energy regime.

It has also been predicted that semiconducting MXenes (e.g., Y_2_CCl_2_, Sc_2_CCl_2_, Y_2_CF_2_, Sc_2_CF_2_, Zr_2_CO_2_, and Hf_2_CO_2_) exhibit high efficiency in the utilization of solar energy, which is comparable to the MoS_2_ monolayer (<3.1 eV). Although the light absorption of Sc_2_CO_2_ is poor, Ti_2_CO_2_ has a wide range of absorption in E < 3.1 eV regions. More to the point, it has been reported that MXenes have excellent mechanical flexibility under strain, and that the bandgap of these structures can be tuned by applying strain, which develops their applications in optoelectronic devices (such as Sc_2_CO_2_, Ti_2_CO_2_, and Zr_2_CO_2_ possessing indirect-to-direct bandgap transition at the strain of 2%, 3%, and 8%, respectively) [157,158].

#### 4.3.3. Thermoelectric Properties

MXenes have been introduced as promising thermoelectric materials for energy conversion applications at high temperatures due to their inherent ceramic nature. In general, it has been predicted that the metallic MXenes have inferior thermoelectric materials while semiconducting MXenes are suitable as thermoelectric materials [159]. For example, Nb_2_CF_2_, as an MXene with metallic behavior, displays an inferior thermoelectric power factor while the semiconducting MXenes (e.g., Mo_2_CF_2_) exhibit comparatively more significant thermoelectric power factor than metallic MXenes. It is recognized that the significant thermoelectric power factor of Mo_2_CF_2_ is because of merging the flat and dispersive bands close to the band edge. Indeed, this type of electronic band structure provides both high electrical conductivity and a significant Seebeck coefficient at low carrier concentrations [160]. In addition, the thermoelectric properties of Mo-based MXens (Mo_2_CT*_x_*, Mo_2_TiCT*_x_*, and Mo_2_Ti_2_C_3_T*_x_*) have been measured [161]. Among these MXenes, Mo_2_TiC_2_T*_x_* has been introduced as the Mo-based MXens with the most significant thermoelectric power factor, which is 3.09 × 10^−4^ W m^−2^ K^−2^ at 803 K. The superior thermoelectric power factor of Mo_2_TiC_2_T*_x_* is attributed to its electronic band structure, including much more flat bands than Mo_2_CO_2_ and Mo_2_Ti_2_C_3_O_2_ near the Fermi level.

## 5. MXenes in Solar Cells

With increasing attention shedding light on the promising future of MXenes in solar cell applications, the roles of MXenes in solar cell structures have become the most challenging issue to inhibit MXenes from practical application. To comprehensively understand the physics/chemistry behind and obtain a better insight on the MXene-based materials for solar cell applications, we categorize the roles of MXenes in solar cells, including an additive in perovskite solar cells, electrodes, hole/electron transport layer, and MXene-silicon-based heterojunction solar cells. Regarding MXenes as an additive in perovskite solar cells, the electron transfer speed and crystal size are significantly accelerated and enlarged, respectively, which have positive contributions to the power conversion efficiency. Meanwhile, introducing MXenes into perovskite solar cells, without affecting other intrinsic properties of perovskite nanoflakes, the work function can be modified effectively, being beneficial to realize high-performance solar cells with high power conversion efficiency and environmental stability. The high conductivity, high transparency, outstanding flexibility, and tunable work function enable MXenes to be a great potential as electrodes in solar cell applications. MXenes can be employed as a back electrode, common electrode, and flexible hybrid electrodes in various types of solar cells. Compared to carbon and ITO electrodes, MXene-based electrodes may noticeably facilitate hole injection rate, reduced series resistance with improving the power conversion efficiency, and enhance the open-circuit voltage. In terms of hole/electron transport layer, due to the tunable work function, the electron potential barrier, interface properties, electron mobility, synergetic effects, and hole transport/collection are significantly enhanced at the interface between aim solar cell channel materials wherein power conversion efficiency, stability, and open-circuit voltage can largely be improved. For MXene-silicon-based heterojunction solar cells, MXenes are employed as a transparent conducting electrode which is a transparent conducting film for charge transport and contributes to the built-in potential to the separation of electron–hole pairs. More importantly, owing to the tunable work function of MXenes, the efficiency of charge separation at a solar cell junction formed between the MXene and Silicon can be significantly improved. Later in this review, the key photovoltaic parameters of solar cells are summarized according to the roles played by the MXenes in solar cells, such as stability, power conversion efficiency, fill factor, current density, and open-circuit voltage, along with the key improvement. Further, those mechanisms enhancing the performance of MXenes in solar cells are deeply discussed for the further development and commercial applications of MXene-based materials in solar cell applications.

### 5.1. Perovskite-Based Solar Cells

PVSK solar cells have witnessed a rapid development during the past several years owing to their light-harvesting properties and many milestones, such as high PCE up to 23.2%, long-term stability over thousand hours, and so forth have been reached in this field. However, to achieve its theoretical PCE limitation (30–33%), a number of challenging issues still needs to be resolved, in particular, the small crystal size, being proportional to the PCE due to the larger crystal size, and the fewer amount of grain boundaries, to name a few. For the first time, Guo et al. proposed employing 2D MXene (Ti_3_C_2_T_x_) as an additive in PVSK solar cells [125,126,127,128,162]. The termination groups of Ti_3_C_2_T_x_ can effectively retard the nucleation process of CH_3_NH_3_PbI_3_, resulting in an enlarged behavior of the crystal size with the consequent enhanced PCE, as illustrated in Figure 17a; the PCE is 1.62% higher than that of PVSK solar cells without Ti_3_C_2_T_x_ MXene. Additionally, via introducing the 0.03 wt.% Ti_3_C_2_T_x_ MXene additive, the electron transfer has been significantly accelerated through the grain boundary as well, and is further proved by the smaller charge transfer resistance of the Ti_3_C_2_T_x_-added PVSK solar cell, as shown in Figure 17b. Interface engineering has been considered as a determining role in PVSK solar cell applications, providing a higher charge collection or injection rate at the interfaces of PVSK/charge transport layers. Among these interface engineering means, inserting a 2D material layer into the PVSK solar cells as a buffer or an interlayer is one of the most effective routes, modifying the work function and the bandgap of the functional materials in PVSK solar cells. Intrigued by this, Agresti et al. demonstrated high-performance PVSK solar cells based on Ti_3_C_2_T_x_ MXene modified PVSK [163]. As shown in Figure 17c and d, they introduced Ti_3_C_2_T_x_ MXene into PVSK as an additive and observed that the work function of the PVSK functional flakes could be effectively suppressed from 4.72 to 4.37 eV without sacrificing other electronic properties, which has a positive contribution to the performance of the device; the PCE exhibited a 26.5% improvement than those without Ti_3_C_2_T_x_. Moreover, the long-term stability of PVSK solar cells can also be significantly improved by inserting Ti_3_C_2_-MXene into CsPbBr_3_ as an interlayer [164]. The inserted Ti_3_C_2_-MXene possesses a work function of 4.5 eV, being higher than the employed carbon electrode (i.e., 4.3 eV), lowering the valence and conduction bands of PVSK flakes, reducing the recombination rate at the interface, and finally resulting in acceleration to the hole transfer and enhancement of photocurrent as depicted in Figure 17e. Further, the inserted Ti_3_C_2_-MXene layer can effectively passivate the surface of PVSK flakes to provide a direct conducting channel between Ti_3_C_2_-MXene and CsPbBr_3_, facilitating the carrier transport speed to the carbon electrode. The stability measurement was performed to evaluate the long-term stability of the device wherein it was seen that under the ambient and high-temperature conditions, the device exhibits excellent operating stability over 1900 h, indicating that Ti_3_C_2_-MXene holds great potential in enhancing the performance of the PVSK solar cells as shown in Figure 17f. To further improve the long-term operation stability, 2D Ruddlesden–Popper PVSK solar cells have recently been proposed. As shown in Figure 17j, Jin et al. [165] reported PVSK solar cells with Ti_3_C_2_T_X_ MXene-doped PVSK flakes; owing to high electrical mobility and conductivity, the current density of the device has been almost linear to the incident light intensity (Figure 17k), suggesting a lower bimolecular recombination degree caused by the reduction in the trap density via introducing MXene as a dopant. Figure 17l demonstrates the stability of the device wherein it is seen that with Ti_3_C_2_T_X_ doped PVSK flakes, the device environmental stability is much better than that of pure PVSK flakes, mostly attributing to a better passivation effect and the crystallinity of PVSK flakes. Apart from the above-mentioned 2D MXene as an additive in PVSK solar cell applications, it has also been employed as conductive bonded bridges in ETLs. Huang et al. fabricated high-performance PVSK solar cells based on multi-dimensional TiO_2_, SnO_2_, and Ti_3_C_2_T_X_ MXene heterojunction [166]. Due to the matched energy alignment of (CH_3_NH_3_)PbI_3_ and FTO (Figure 17g), the hole recombination at the interface between the FTO layer and ETL has efficiently been suppressed, resulting in a higher PCE. Moreover, owing to the unique structure of the multi-dimensional conductive network below the ETL; the moisture-resistance ability has also been significantly improved (more than 45 days) than that of pure SnO_2_ ETL, as presented in Figure 17h. The in situ solution growth method is also performed to form MXene-MAPbBr_3_ heterojunction. Owing to the matched energy levels, the charge- and energy-transfer speed at MXene-MAPbBr_3_ heterojunction are significantly facilitated, having a positive contribution to the performance enhancement, as shown in Figure 17i [167].

### 5.2. Electrodes

As said earlier, the tunable work function, high transparency, electrical conductivity, and excellent flexibility of MXene, enable this fascinating 2D material to hold great potential for optoelectronic device applications, in particular, as an electrode for solar cell applications [168]. In this section, we highlight some recent representative progress of MXene electrodes applications with a sequence of dye-sensitized, PVSK, quantum dot-sensitized, and flexible solar cells.

For DSSCs, TiO_2_ is commonly used as a mesoporous semiconductor electrode; however, thick TiO_2_ electrodes are difficult to fabricate since the pore size is unproportioned to the film’s mechanical strength. Thus, Dall’Agnese et al. proposed a coating of MXene nanoflakes on FTO transparent substrates to form porous TiO_2_ electrodes rather than using a paste containing NPs [169]. The measured oxidation temperature dependence on the incident PCE (IPCE) indicates that the highest IPCE could be achieved at 450 ℃ (Figure 18a). When the temperature is changed, the IPCE presents a decreasing behavior, which is mainly caused by the combination of the smaller surface area and the absence of semiconducting TiO_2_ particles. To obtain better insight of the MXene oxidized porous TiO_2_ electrode-based DSSCs, the open-circuit voltages (*V_oc_*) versus electron density was investigated; as shown in Figure 18b, the measured *V_oc_* was almost 300 mV smaller than that of DSSCs without MXene oxidized porous TiO_2_ electrode, mostly attributed to the large trap density during the MXene oxidizing process. Although the outcomes still are not comparable to that of commercially available TiO_2_ electrode-based DSSCs, the proposed method is promising in regards to the fabrication of high-performance oxide films for DSSCs applications. For PVSK solar cell applications, Cao et al. [170] fabricated noble-metal-free PVSK solar cells based on 2D Ti_3_C_2_ electrode via a hot-pressing method and observed that the work function of employed Ti_3_C_2_ is well-matched with the valence band of employed (CH_3_NH_3_)PbI_3_, i.e., both the electrons and holes can be extracted from (CH_3_NH_3_)PbI_3_ layer successfully, having a positive contribution to the PCE as shown in Figure 18c (27% higher than that of the electrode without Ti_3_C_2_). Moreover, Figure 18d demonstrates the employed Ti_3_C_2_ layer acting as an encapsulating layer to improve the environmental stability two times greater than that of the electrode without Ti_3_C_2_, under ambient conditions. Subsequently, Mi et al. demonstrated a high-efficiency PVSK solar cell based on carbon-CNTs-MXene mixed dimensional electrode [171]. The electrode conductivity and surface contact with carbon paste can significantly be enhanced via the mixed dimensional electrode, leading to a champion PCE as high as 7.09%. The stability under moisture can be maintained over one month owing to the seamless interfacial contact of each layer, as indicated in Figure 18f. Other than the abovementioned two types of solar cells, quantum dot sensitized solar cells (QDSCs) are another promising third-generation solar cell owing to their inexpensive materials, simple fabrication processes, as well as high theoretical efficiencies. MXene is a promising counter electrode (CE) candidate due to its electrical conductivity and electro-catalytic activity. Chen et al. presented QDSCs based on CuSe/Ti_3_C_2_ composite CE synthesized via a one-step hydrothermal method [172]. Electrochemical impedance spectroscopy (EIS) was performed to evaluate the electro-catalytic activity of employed CEs with various Ti_3_C_2_ concentrations. As indicated in Figure 18e, the CuSe/Ti_3_C_2_-30 mg CE possesses the smallest sheet resistance (1.29 Ω), suggesting the intrinsic electrical conductivity of CuSe to be improved by the applied Ti_3_C_2_ being mainly due to the increased number of catalytically active sites and unique three-dimensional structure. These outstanding findings provide a novel route to realizing high-performance QDSCs based on MXene composite CEs. Subsequently, Tian et al. demonstrated QDSCs based on CuS/Ti_3_C_2_ CE prepared via a facile ion-exchange method with the preparation procedure presented in Figure 18g [173]. Thanks to the advantages provided by Ti_3_C_2_ (e.g., high electrical conductivity) and CuS NPs (e.g., abundant catalytically active sites), the IPCE of the device has been measured to be 1.5 times higher than that of QDSCs with pure CuS CE and with a value up to 5.11%, as shown in Figure 18h. As aforementioned, MXene is a promising candidate for flexible solar cell applications. Integrated with Ag nanowires, the Ag/MXene hybrid can be a flexible and transparent electrode, and can be produced via a scalable and simple solution-processed method for high efficiency flexible organic solar cells (Figure 18i) [174]. Owing to the superior properties provided by flexible and transparent Ag/MXene electrodes, including high electrical conductivity, low roughness, and high transmittance, the PCE of the device was as high as 8.30%. Meanwhile, since the robust mechanical performance of the employed Ag/MXene electrode, the performance was significantly promoted even after 1000 bending (5 mm bending radius) and unbending cycles treatment (Figure 18j). Qin et al. presented a flexible photovoltaic supercapacitor (PSC) by utilizing Ti_3_C_2_T_x_ MXene flake as an electrode, which is prepared via an all-solution-processed route and with a champion PCE of 13.6%, which is competitive with that of ITO electrode-based QDSCs, as shown in Figure 18k [175]. Meanwhile, the storage efficiency and average transmittances were 88% and 33.5%, respectively, suggesting Ti_3_C_2_T_x_ MXene flake as a suitable material for cost-efficient, flexible, miniature and wearable electronic device applications.

### 5.3. Hole/Electron Transport Layer (HTL/ETL)

The tunable work function of MXene enables this fascinating novel material to serve as HTL/ETL to further promote the performance of various types of solar cells. In the present section, we summarize the recent progress of MXenes acting as HTL/ETL in solar cell applications.

To substitute PEDOT:PSS HTL, Hou et al. employed Ti_3_C_2_T_x_ flakes as HTL in PBDB-T:ITIC solar cell, as indicated in Figure 19a [177]. According to Figure 19b, the hole transmission was facilitated due to the matched energy alignment of Ti_3_C_2_T_x_ and ITO; the PCE of the device was enhanced with a champion value of 10.53%, which is significantly higher than that of the pure ITO-based device (i.e., 4.21%). Furthermore, the inserted Ti_3_C_2_T_x_ HTL may efficiently suppress the corrosion of PEDOT:PSS toward active layers and ITO substrates, thus, enhance the long-term stability of the device, as shown in Figure 19c. Liu et al. employed Mo_1.33_C MXene-assisted PEDOT:PSS as HTL to realize high-performance polymer solar cells [178]; the architecture of the device is shown in Figure 19d. The external quantum efficiency (EQE) and short-circuit current density (*J_sc_*) measurements were performed to evaluate the device’s performance. As shown in Figure 19e, compared to the device with PEDOT:PSS HTL, both the EQE and *J_sc_* were enhanced wherein the Mo_1.33_C MXene-assisted PEDOT:PSS HTL may strengthen the charge extraction/transport efficiencies, suppress the series resistance, and finally improve the photovoltaic performance of the device. Similarly, Hou et al. utilized PEDOT:PSS/Ti_3_C_2_T_X_ MXene composite layers as HTL to achieve high-performance polymer solar cells [179]. The configuration of the device is shown in Figure 19f. As indicated in Figure 19g, the employed MXene flake is favorable to form a matched energy-level alignment between ITO and active layer, thus facilitating hole transport from the active layer to ITO and yielding outstanding photovoltaic performance, i.e., the PCE increasing from 13.10% to 14.55% with PEDOT:PSS/Ti_3_C_2_T_X_ MXene HTL (Figure 19h). Furthermore, the corrosion of PEDOT:PSS toward the photoactive layer and ITO substrate was suppressed via the inserted MXene-based HTL, leading to an improvement in the long-term stability over 300 h.

Apart from the HTL application, MXene has also been employed as ETL to promote the performance of solar cells [180,181]. Recently, Yang et al. used Ti_3_C_2_T_X_ MXene nanoflakes as an ETL for PVSK solar cells via spin-coating process; the cross-sectional SEM image of the device is shown in Figure 20a [182]. Applying an ultraviolet–ozone treatment at different times, the Fermi level of the applied Ti_3_C_2_T_X_ can be modified from −5.52 to −5.62 eV (see Figure 20b). Meanwhile, the oxide-like Ti–O bonds were generated during the ultraviolet–ozone process, which is beneficial for improving the interface properties between MAPbI_3_ and Ti_3_C_2_T_X_. Remarkably, the highest PCE of 17.17% was achieved with 30 min ultraviolet–ozone treatment, which is mainly due to the downshifted Fermi level of employed Ti_3_C_2_T_X_ MXene, having a positive contribution for the electron transportation. The long-term stability measurement was carried out as well, as shown in Figure 20c, with 30 min ultraviolet-ozone treatment, the device maintained 70% of the initial PCE over 800 h, which is nearly 500 h longer than that of the device without ultraviolet–ozone treatment. It is noted that the underlying mechanism(s) needs to be further investigated. Compared to pristine MXene ETL, it can be utilized as a conductive additive to form ETL with other materials, providing many advantageous properties. Intrigued by this, Yang et al. demonstrated PVSK solar cells with SnO_2_–Ti_3_C_2_ MXene nanocomposites ETL; the cross-sectional SEM image of the device is shown in Figure 20d [183]. The inserted SnO_2_–Ti_3_C_2_ MXene layer is an effective agent to form a matched energy-level alignment between ITO and CH_3_NH_3_PbI_3_ layer and to increase the conductivity of TEL, leading to an enhanced PCE from 17.23% to 18.34% with SnO_2_–Ti_3_C_2_ (1.0 wt% Ti_3_C_2_) ETL. Furthermore, the measured EQE and integrated current densities of devices with SnO_2_, SnO_2_–Ti_3_C_2_ (1.0 wt% Ti_3_C_2_) and Ti_3_C_2_ further confirmed the great potential of MXenes toward high-performance solar cell applications (see Figure 20e,f). Soon after, Wang et al. reported PVSK solar cells with a champion PCE of 20.65% with MXene modulated SnO_2_ ETL, the cross-sectional SEM image of the device is shown in Figure 20g [184]. Combined with a density functional theory (DFT) simulation, the efficient charge transfer, and enhanced electron mobility were attributed to the energy-level alignment caused by the applied MXene modulated SnO_2_ ETL (Figure 20h). Furthermore, a stable PCE of 20.47% and an output current density of 21.81 mA·cm^−2^ were achieved with 0.5 mg·mL^−1^ MXene modulated SnO_2_ ETL, as shown in Figure 20i. Thanks to the growth of high-quality PVSK flake, the FTO/MXene/SnO_2_ surface has a smoother and more hydrophobic morphology.

To overcome the lead toxicity and instability of the APbX3 structure, an inorganic lead-free Cs_2_AgBiBr_6_ double PVSK structure may be considered as a promising candidate for solar cell applications; however, the low crystallization severely hindered its development towards being a high PCE solar cell. Motived by this, Li et al. proposed a novel and simple strategy, utilizing monolayer MXene nanoflakes doping TiO_2_ (Ti_3_C_2_T_x_@TiO_2_) as ETL to fabricate efficient double PVSK solar cells; the architecture of the device is shown in Figure 21a [185]. Owing to the superior conductivity of Ti_3_C_2_T_x_@TiO_2_ ETL, the electron accumulation and hole transfer at the interface between Ti_3_C_2_T_x_@TiO_2_ and Cs_2_AgBiBr_6_ were significantly suppressed, leading to a high PCE compared to that of TiO_2_ ETL, as shown in Figure 21b. Moreover, the stability of the device can be maintained for 15 days with Ti_3_C_2_T_x_@TiO_2_ ETL under ambient conditions without any encapsulation, emphasizing the suitability of monolayer MXene as a material to improve the photostability of double PVSK solar cells. The oxidation of Ti_3_C_2_T_x_ hydrocolloid was also employed as an ETL to modify the performance of PVSK solar cells [186]. The schematic architecture and cross-sectional SEM image of the device are shown in Figure 21c,d, respectively. Through a DFT simulation, the energy levels of each layer were calculated, as indicated in Figure 21e; after a heavy oxidation process, a transformation from metallic to semiconducting behavior occurred, resulting in an appropriate energy level alignment suppressing the electron–hole recombination rate at the interface between PVSK and ETLs. The HO-Ti_3_C_2_T_x_ ETL exhibited a larger *J_sc_* than the other ETLs (Figure 21f), indicating that the HO-Ti_3_C_2_T_x_ ETL holds a superior electron mobility, being suitable for high PCE PVSK solar cell applications. Other than PVSK solar cells, MXene ETL has been applied to realize high-performance polymer solar cells. Hou et al. studied the fullerene and non-fullerene polymer solar cells with ZnO/Ti_3_C_2_T_x_ composite ETL [187]. The applied Ti_3_C_2_T_x_ nanoflakes provide additional electron transport pathways within the ZnO NPs (Figure 21g). Moreover, the average PCE of the device with ZnO/Ti_3_C_2_T_x_ ETL was 12.20%, which is 15.53% higher than that of pristine a ZnO ETL-based device, which can be attributed to the following two main factors: (i) Ti_3_C_2_T_x_ nanoflakes formed 3D interconnected bridges amongst ZnO layers and hence, the charge transfer between the ZnO NPs are significant; (ii) the surface of ZnO is passivated by the formed Zn–O–Ti bonding, which can prolong the exciton lifetime so that electron may transfer across the Zn–O–Ti bonding. Compared to the device with pristine ZnO ETL, the stability of ZnO/Ti_3_C_2_T_x_ composite ETL-based device was efficiently promoted, i.e., retaining 72.46% of its initial PCE over 40 days (Figure 21h), indicating the ZnO/Ti_3_C_2_T_x_ composite ETL can improve the crystallinity and micromorphology of employed polymer materials. The mentioned outcomes further confirm MXene as a suitable candidate for high-performance energy harvesting device applications.

### 5.4. MXene-Silicon-Based Heterojunction

Nowadays, silicon-based solar cells are the most frequently used energy harvesting devices. To further promote the performance, 2D materials such as MXenes can be employed owing to their unique structures and advantageous optoelectronic properties. MXenes, as a new member of the 2D material family, have widely been investigated owing to their excellent optoelectronic properties. Recently, MXene-silicon-based heterojunctions have attracted considerable attention in solar cell technology due to the tunable work function, high electrical conductivity provided by MXene, etc. Motived by this, Fu et al. fabricated an *n^+^–n–p^+^* Si-based solar cell on Ti_3_C_2_T_x_ MXene back electrode; the device architecture is shown in Figure 22a [188]. The formed ohmic contact between *n^+^–*Si and applied Ti_3_C_2_T_x_ MXene promotes the electron transfer and suppresses the electron–hole recombination rate, leading to a high *J_sc_* and *V_oc_*. Meanwhile, within the rapid thermal annealing treatment, the series resistances decreased monotonously as the temperature increased from 100 ℃ to 400 ℃ (Figure 22b), indicating the physical adhesion and electrical contact of the MXene- *n^+^–*Si heterojunction could be further enhanced. Yu et al. studied Ti_3_C_2_T_x_-on-Si heterojunction-based silicon solar cells [189] wherein Ti_3_C_2_T_x_ served as both the component to establish the Schottky junction with *n^+^–*Si and the electrode for hole collection between the transparent conducting electrode and Si surface, as shown in Figure 22c. Noticeably, the PCE of the device can be further improved over 9.0% via introducing two-step chemical treatments (2.0% AuCl_3_ and HCl) and over 10% by inserting an antireflection layer, as shown in Figure 22d. These enhancements are mainly attributed to (i) the doping effect introduced by HCl, (ii) the increased Schottky barrier height, and (iii) promoted charge transfer originated from Au NPs from the applied AuCl_3_. This novel Ti_3_C_2_T_x_-on-Si heterojunction-based silicon solar cell may provide a promising means to realize high-performance solar cells based on MXene-silicon heterojunction.

## 6. Summary and Future Perspectives

MXenes are a class of two-dimensional material that have a wide variety of applications ranging from sensing to advanced energy systems owing to exceptional tailor-made properties. This family of 2D materials has rich chemistry and exceptional electrical, thermoelectrical, optical, and mechanical properties, making these novel materials a promising candidate for the next generation of energy materials. Very recently, MXene has started its way in those materials and structures employed in solar cell applications. The purpose of this mini review was to comprehensively discuss the basics and outline the opportunities available in this growing field, starting with 2D material candidates for solar cell applications to compare and highlight the advantages and disadvantages of all 2D materials, including MXenes. The common manufacturing techniques that are usually utilized to fabricate the solar cell materials are then deeply elaborated to see what kind of process/method is suitable for specific materials in solar cell technologies. After providing a comprehensive discussion regarding the detailed synthesis processes as well as the key properties needed in solar materials, we then provided a critical review and outline of recent research efforts investigating the application of MXene-based materials in solar cells. As the present topic is in its infancy, it is believed MXene will find its foothold in the near future and in this ever-growing domain of nanoscience. Although MXenes are considered to be a promising candidate for high-performance or novel solar cell applications, challenges and opportunities still remain for researchers.

For solar cells to employ MXene as an additive: (1) the long-term stability and PCE should be further improved; (2) to obtain a better insight into the enhancing mechanisms provided by MXene, the Fermi level and the properties of various functional groups of MXenes should be further investigated through the combination of both theoretical simulations and experimental data; (3) to meet the demands of various applications, MXene-based novel energy harvesting devices, such as flexible solar cells, supercapacitors, and so forth, should be developed; (4) the environmental stability of MXene-based solar cells should be further enhanced via introducing passivation and encapsulation processes; (5) other than commonly used Ti_3_C_2_T_x_ MXene, other kinds of MXene suitable for solar cell applications should be explored further; (6) MXene-based solar cells with novel structures, in particular, enhance light–matter interaction and accelerate carrier transport behavior. In summary, MXene is a fascinating candidate for solar cell applications owing to its extraordinary properties; however, still, a number of serious challenges should be addressed in order to develop high-performance solar materials.

## Figures and Tables

**Figure 1 nanomaterials-11-03170-f001:**
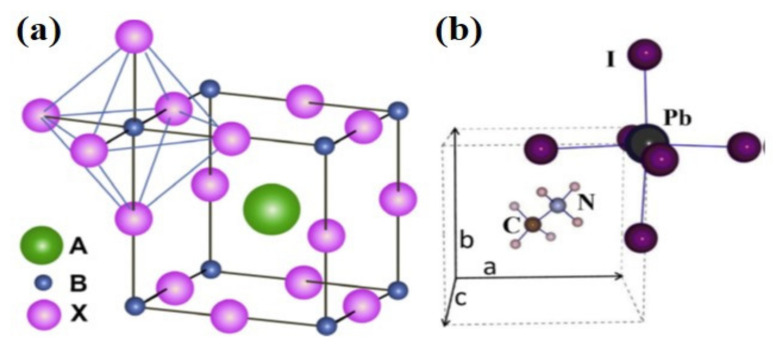
Perovskite structure with the general chemical formula of ABX_3_: (**a**) A and B represent positively charged metals and red spheres anion atoms; (**b**) the central A site filled with Methylammonium cation (${\mathrm{CH}}_{3}{\mathrm{NH}}_{3}^{+}$). Reproduced with permission from Reference [5]. Copyright 2015, Elsevier Ltd.

**Figure 2 nanomaterials-11-03170-f002:**
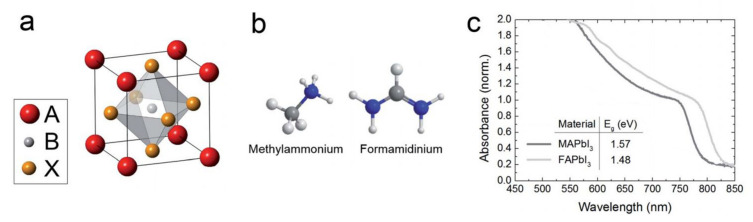
Effect of A cation on the perovskite bandgap: (**a**) the ABX_3_ perovskite crystal structure; (**b**) the atomic structure of methylammonium (MA) and formamidinium (FA) as A-site cations; (**c**) UV-Vis spectra for the APbI_3_ perovskites, where A is either MA or FA. Reproduced with permission from Reference [9]. Copyright 2011, Royal Society of Chemistry.

**Figure 3 nanomaterials-11-03170-f003:**
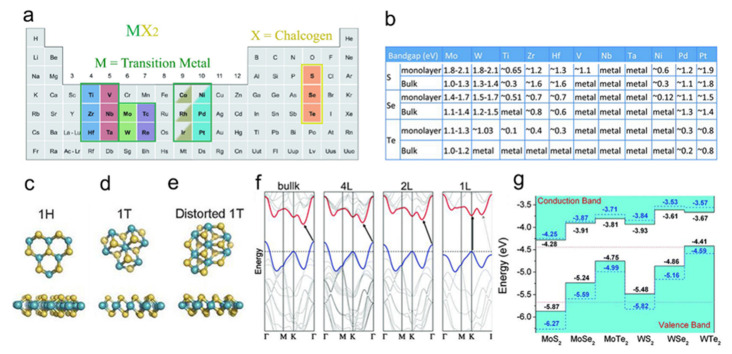
Different stable phases of 2D TMDs and their electronic properties: (**a**) periodic table showing elements used to build TMDs consisting of 16 transition metals and three chalcogen atoms; (**b**) the electronic properties and bandgap of the common compounds of the family of TMDs; (**c**) 1H phase; (**d**) ideal (a × a) 1T phase; (**e**) distorted (2a × a) 1T phase. It is illustrated that a single layer of the transition metal atoms (blue) is sandwiched between the two layers of chalcogen atoms (yellow); (**f**) band structure quadrilayer (4L), bilayer (2L), and monolayer (1L) MoS_2_, from left to right, illustrating the indirect-to-direct bandgap transitions following by a widening of the bandgap. The conduction and valence band edges are highlighted with red and blue lines, respectively. (**g**) The relative conduction and valence band edges of the semiconducting monolayer TMDs. Panel a reproduced from Reference [25], Copyright 2013 Nature Publishing Group. Panel b was taken from Reference [28] and panels c, d, and e were reproduced from Reference [29], Copyright 2015 Royal Society of Chemistry. Panel f was taken from Reference [30], Copyright 2010 American Chemical Society. Panel g was taken from Reference [31], Copyright 2013 AIP Publishing LLC.

**Figure 4 nanomaterials-11-03170-f004:**
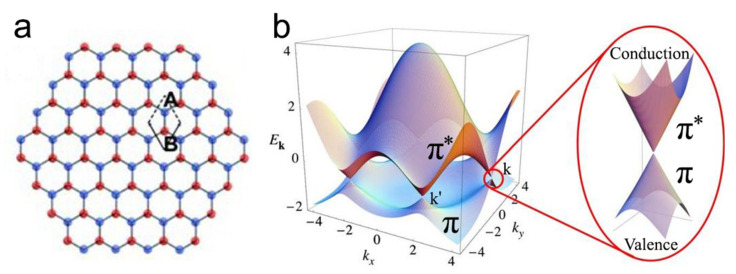
Structure and electronic band structure of graphene: (**a**) hexagonal honeycomb lattice of graphene including two atoms (A and B) in a unit cell; (**b**) electronic dispersion in the honeycomb lattice, in which the conduction band and valence band meet at six Dirac points (left), and close view of the energy bands close to one of the Dirac points (right). Reproduced with permission, Panel a taken from Reference [49], Copyright 2010 American Chemical Society, Panel b from Reference [50], Copyright 2009 American Physical Society.

**Figure 5 nanomaterials-11-03170-f005:**
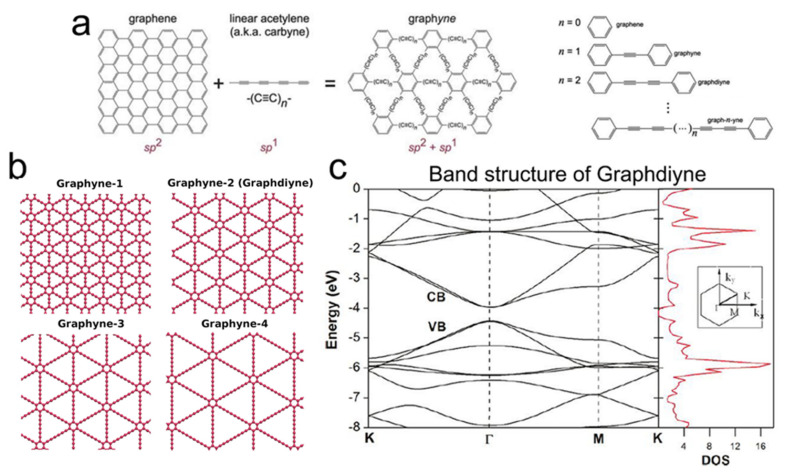
Structure of the single-layer graphynes and band structure of graphdiyne: (**a**) schematic structures of graphene and graphynes, where two neighboring sp^2^-hybridized carbon atoms in graphene are interlinked by *n* “–C≡C–” bonds to form graphyne-*n*; (**b**) graphyne-*n* structures (*n* = 1–4). Graphyne-2 is called graphdiyne (GDY); (**c**) band structure and density of states for the single-layer graphdiyne sheet. Adopted with permission, Panel a from Reference [68], Copyright 2012 Royal Society of Chemistry. Panel b from Reference [69], Copyright 2016 Nature Publishing. Panel c from Reference [70], Copyright 2011 American Chemical Society.

**Figure 6 nanomaterials-11-03170-f006:**
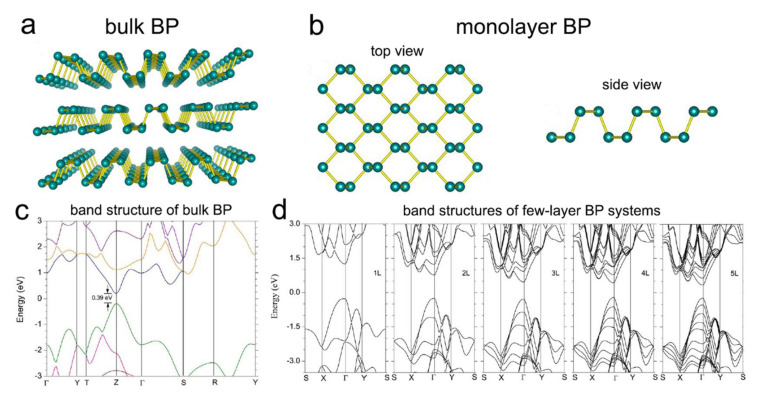
Structure of black phosphorous systems and their band structures: (**a**) structure of bulk BP; (**b**) top and side views of the 2D monolayer BP; (**c**) band structure of the bulk BP; (**d**) band structures of different few-layer BP systems. The direct bandgap decreases as the thickness increases. Panels a and b reproduced from Reference [80], Copyright 2014 IOP Publishing. Panels c and d reproduced from Reference [82], Copyright 2014 Springer Nature Limited.

**Figure 7 nanomaterials-11-03170-f007:**
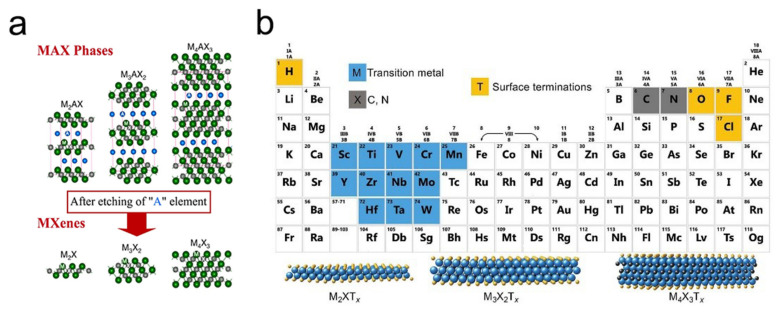
(**a**) Crystal structures of MAX phases and the derived 2D MXenes. (**b**) Periodic table showing elements used to build MXenes. MXenes built with Sc and Mn have not yet been confirmed experimentally. Schematics of three typical MXene structures are presented beneath. Adopted with permission, Panel b is from Reference [92]. Copyright 2019 American Chemical Society.

**Figure 8 nanomaterials-11-03170-f008:**
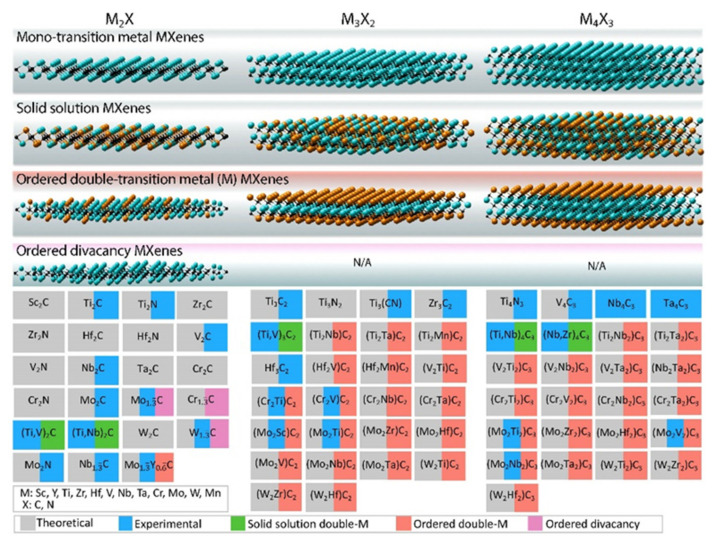
MXene compositions reported to date. The first row shows the structures of mono-M MXenes. The second row illustrates double-M solid solution (SS) MXenes, and their compositions are marked in green. The third row presents ordered double-M MXenes, and their compositions are marked in red. The fourth row shows an ordered divacancy structure, and the relevant compositions are marked in pink. MXene compositions explored experimentally are shown in blue, and those investigated theoretically are marked in gray. Surface terminations are not shown. Adopted with permission from Reference [92]. Copyright 2019 American Chemical Society.

**Figure 9 nanomaterials-11-03170-f009:**
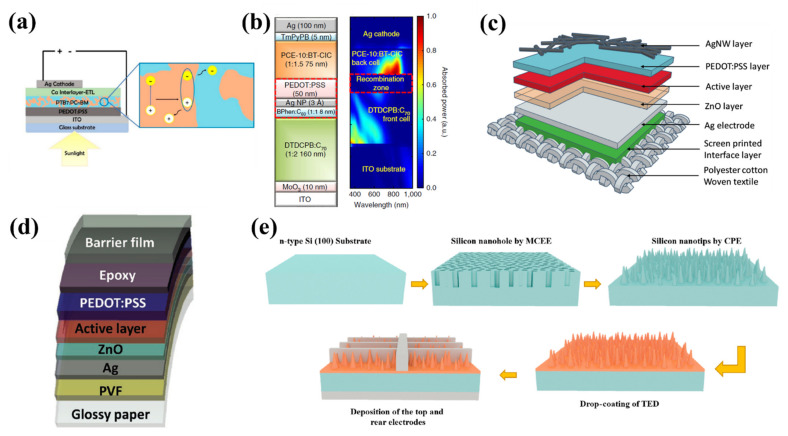
(**a**) Schematic diagram of bulk heterojunction organic solar cell. Reproduced from Reference [108]. Copyright 2019, MDPI. (**b**) Schematic diagram of organic tandem solar cell. Reproduced with permission from Reference [109]. Copyright 2018, Springer Nature. (**c**) Schematic diagram of textile organic solar cells. Reproduced with permission from Reference [110]. Copyright 2019, WILEY-VCH. (**d**) Schematic diagram of paper organic solar cell. Reproduced with permission from Reference [111]. Copyright 2019, WILEY-VCH. (**e**) Schematic diagram of the hybrid solar cell fabrication process. Reproduced with permission from Reference [112]. Copyright 2018, American Chemical Society.

**Figure 10 nanomaterials-11-03170-f010:**
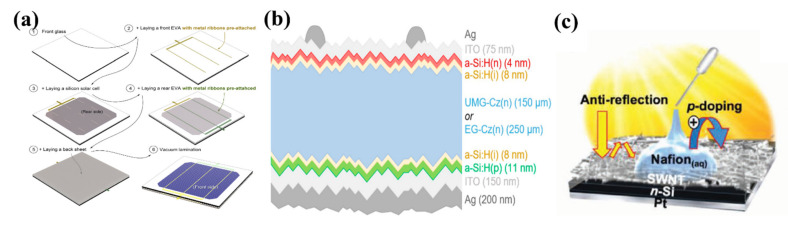
(**a**) Fabrication process of silicon solar cells with 100 μm silicon solar module by LOM method. Reproduced with permission from Reference [113]. Copyright 2021, Elsevier B.V. (**b**) Schematic diagram of silicon heterojunction solar cell. Reproduced with permission from Reference [114]. Copyright 2019, WILEY-VCH. (**c**) Schematic diagram of Nafion-applied single-walled carbon nanotubes and n-type Silicon solar cell. Reproduced with permission from Reference [115]. Copyright 2019, WILEY-VCH.

**Figure 11 nanomaterials-11-03170-f011:**
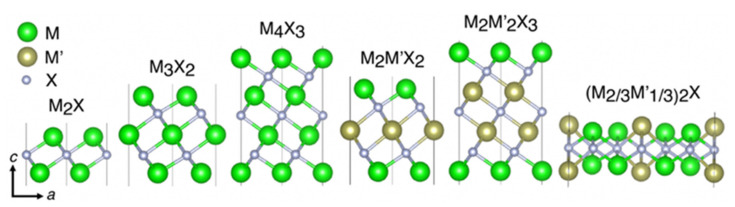
2D MXenes with different structures.

**Figure 12 nanomaterials-11-03170-f012:**
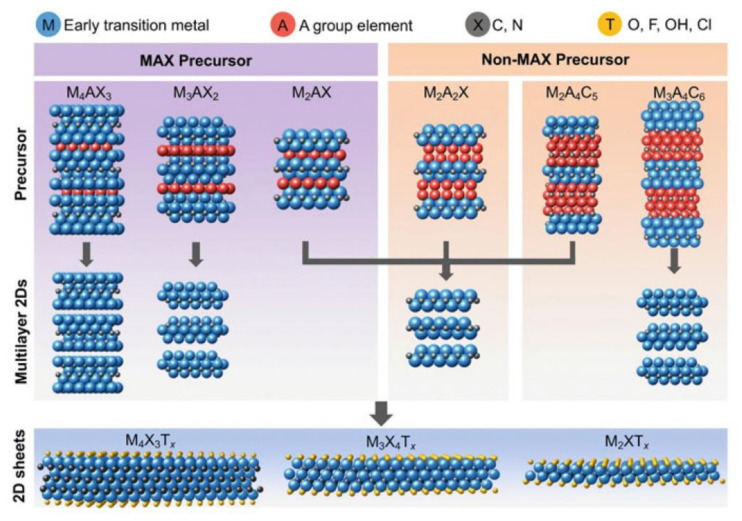
Top-down synthesis approaches of MXenes from their MAX and non-MAX phase precursors. Three different main types of multilayer and monolayer of MXenes are labeled as multilayer 2Ds and 2D sheets. Multilayer 2Ds can be functionalized with surface functions similar to 2D sheets; however, functional groups are not displayed multilayer 2Ds for simplicity. Taken from Reference [134]. Copyright 2019 Springer.

**Figure 13 nanomaterials-11-03170-f013:**
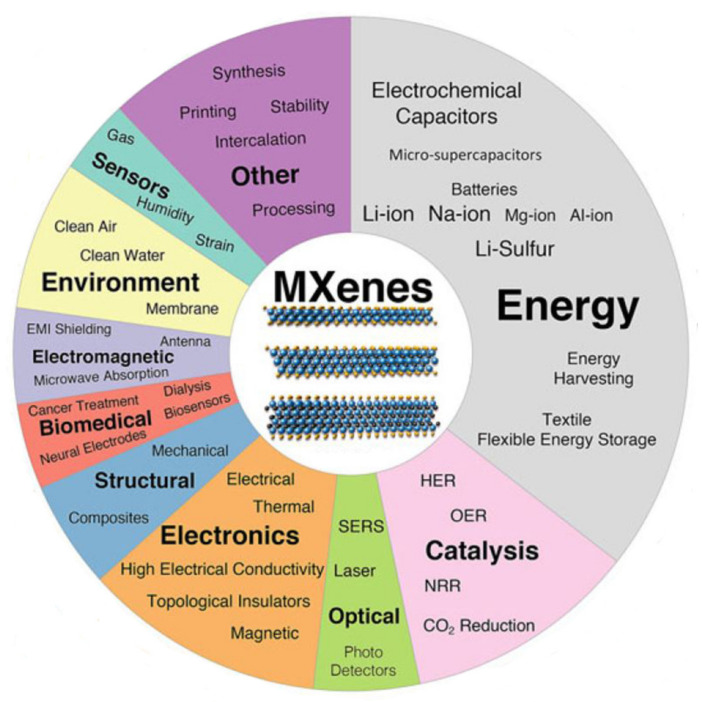
Applications and properties of MXenes. Reproduced from Reference [134]. Copyright 2019 Springer.

**Figure 14 nanomaterials-11-03170-f014:**
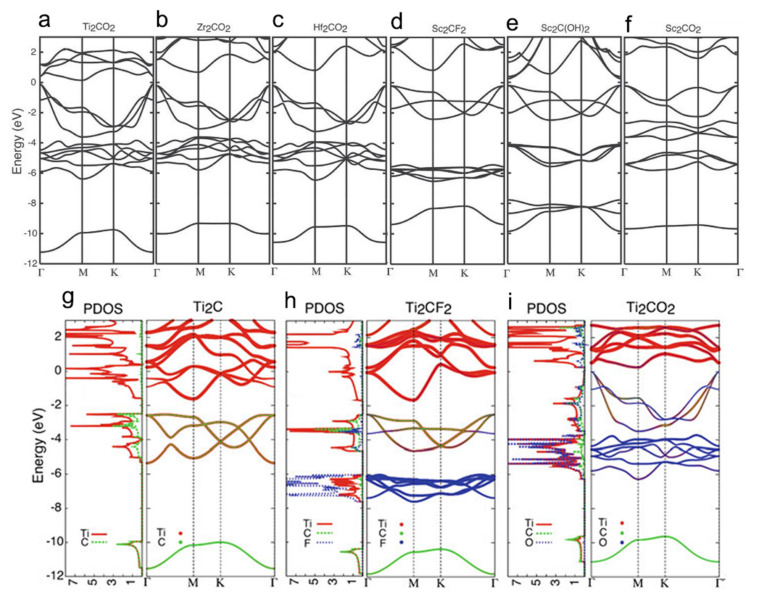
Projected band structures of: (**a**) Ti_2_CO_2_; (**b**) Zr_2_CO_2_; (**c**) Hf_2_CO_2_; (**d**) Sc_2_CF_2_; (**e**) Sc_2_C(OH)_2_; (**f**) Sc_2_CO_2_. Projected density of states band structures of: (**g**) Ti_2_C; (**h**) Ti_2_CF_2_; (**i**) Ti_2_CO_2_. The Fermi energy is located at zero energy. Panels a-i reproduced from Reference [101], Copyright 2013 Wiley-VCH GmbH, Weinheim.

**Figure 15 nanomaterials-11-03170-f015:**
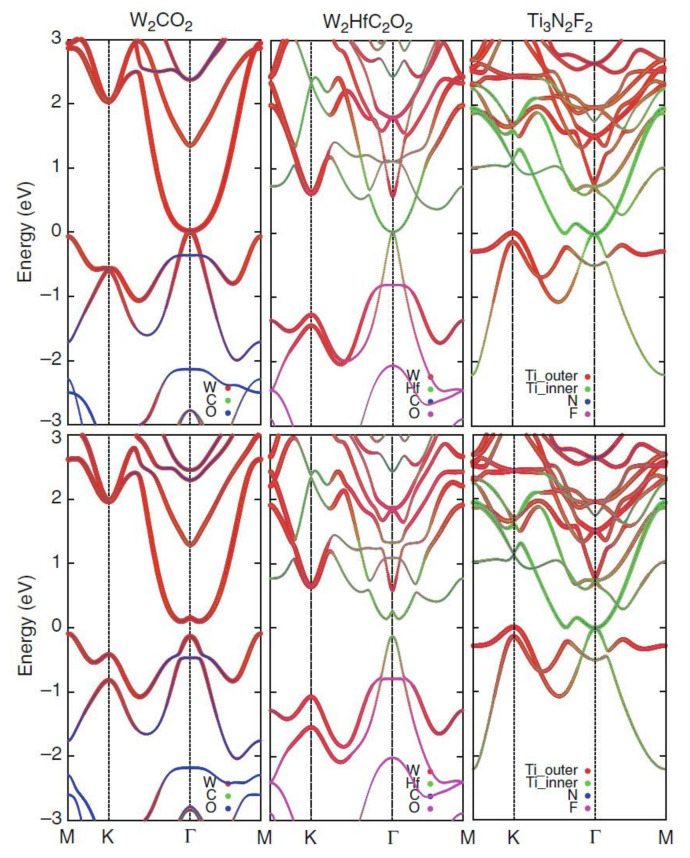
Projected band structures for W_2_CO_2_, W_2_HfC_2_O_2_, and Ti_3_N_2_F_2_. Top (bottom) panels show the band structures calculated without (with) the effect of spin-orbit coupling. The Fermi energy is shifted to zero energy. Taken from Reference [151], Copyright 2016 American Physical Society.

**Figure 16 nanomaterials-11-03170-f016:**
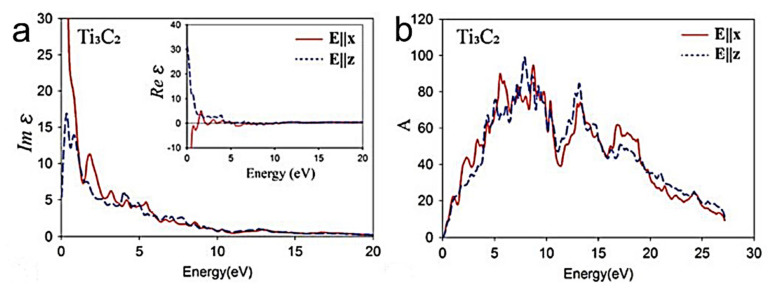
(**a**) The imaginary part of the dielectric function and the real part is shown in the inset. (**b**) The absorption coefficient for Ti_3_C_2_ MXene. The solid red and dashed blue line illustrates the nonzero components of the dielectric *ε*_xx_(ω) and *ε*_zz_(ω) for in-plane (*E*||*z*) and out-of-plane (*E*||*x*) electric field. Panels a and b reproduced from Reference [155], Copyright 2014 Elsevier Ltd.

**Figure 17 nanomaterials-11-03170-f017:**
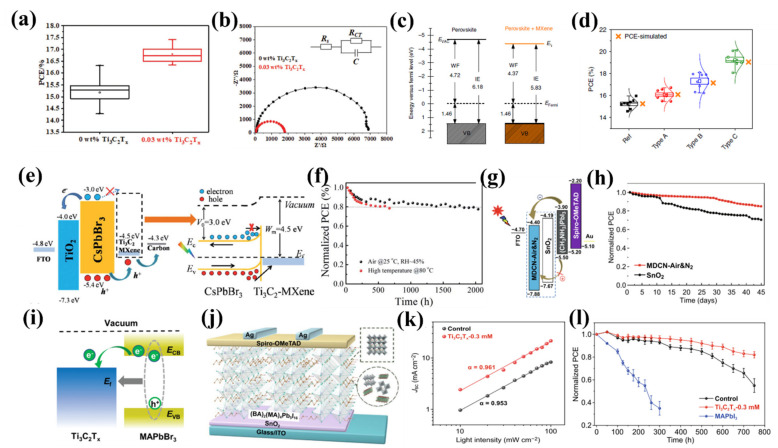
(**a**) The measured PCE of the device with and without 0.03 wt.% Ti_3_C_2_T_x_. (**b**) The Nyquist plots of the device with and without 0.03 wt.% Ti_3_C_2_T_x_ under dark conditions with a 0.7 V bias voltage. Reproduced with permission from Reference [162]. Copyright 2018, Wiley-VCH. (**c**) The energy scheme of perovskite device with and without doped MXene. (**d**) The simulated PCE of three different types of PVSK solar cells. Reproduced with permission from Reference [163]. Copyright 2019, Springer Nature. (**e**) The energy bandgap and carrier transport mechanism of the device under illumination at the interface, respectively. (**f**) The measured PCE of the device as a function of the time under ambient and high-temperature conditions. Reproduced with permission from Reference [164]. Copyright 2019, Royal Society of Chemistry. (**g**) The energy scheme of the device. (**h**) The measured PCE of the device with MDCN-Air&N_2_ and SnO_2_ ETL. Reproduced with permission from Reference [166]. Copyright 2020, Springer Singapore. (**i**) The energy scheme and energy transfer between MXene and perovskite nanoflakes. Reproduced with permission from Reference [167]. Copyright 2020, Wiley-VCH. (**j**) The schematic diagram of the device. (**k**) The Nyquist plots of the device with optimized Ti_3_C_2_T_x_-doping PVSK solar cells. (**l**) The measured PCE of Ti_3_C_2_T_x_-doping PVSK solar cells. Reproduced with permission from Reference [165]. Copyright 2021, Springer Singapore.

**Figure 18 nanomaterials-11-03170-f018:**
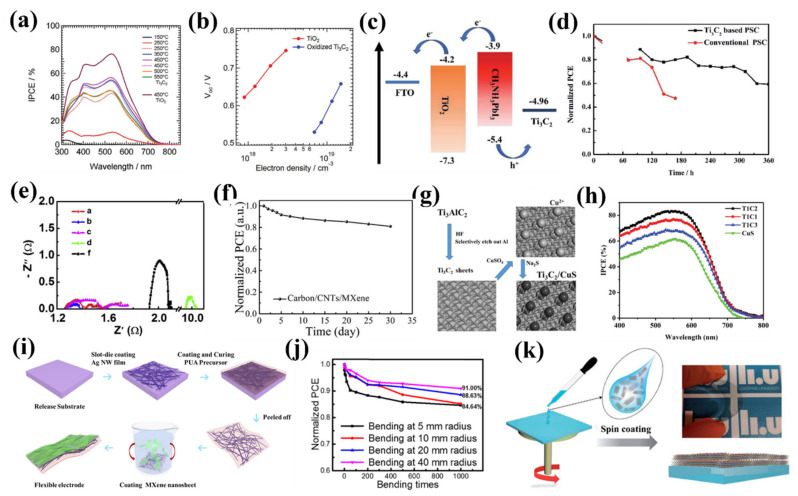
(**a**) The measured IPCE of the device as a function of wavelength at various temperatures. (**b**) The open-circuit voltage and electron density of the device. Reproduced with permission from Reference [169]. Copyright 2018, Royal Society of Chemistry. (**c**) The energy scheme of PVSK solar cells with MXene back electrode. (**d**) The measured PCE of the device under ambient conditions. Reproduced with permission from Reference [170]. Copyright 2019, Royal Society of Chemistry. (**e**) The Nyquist plots of the device, (**a**) CuSe/Ti_3_C_2_-15 mg CE, (**b**) CuSe/Ti_3_C_2_-30 mg CE, (**c**) CuSe/Ti_3_C_2_-60 mg CE, (**d**) CuSe CE, (**f**) Ti_3_C_2_ CE. Reproduced with permission from Reference [172]. Copyright 2019, Elsevier B.V. (**f**) The measured PCE under ambient condition. Reproduced with permission from Reference [171]. Copyright 2020, Royal Society of Chemistry. (**g**) The fabrication process of Ti_3_C_2_/CuS counter electrodes via the ion-exchange method. (**h**) The measured IPCE of the various counter electrodes. Reproduced with permission from Reference [173]. Copyright 2020, Royal Society of Chemistry. (**i**) The fabrication process of MXene-based flexible transparent electrode. (**j**) Normalized PCE of flexible PSCs with MXene/AgNW-PUA transparent electrodes at different bending radiuses versus bending times. Reproduced with permission from Reference [176]. Copyright 2019, American Chemical Society. (**k**) Schematic diagram of transparent flexible electrode fabrication process. Reproduced with permission from Reference [175]. Copyright 2020, Royal Society of Chemistry.

**Figure 19 nanomaterials-11-03170-f019:**
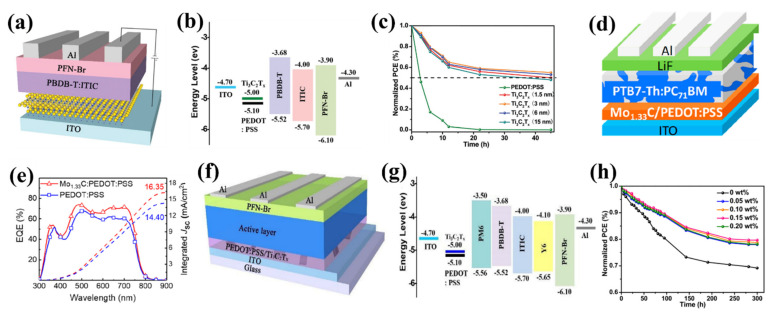
(**a**) Schematic diagram of the device. (**b**) The energy band diagrams of employed materials. (**c**) The stability measurement of the devices with various HTLs in ambient conditions without encapsulation. Reproduced with permission from Reference [177]. Copyright 2019, Royal Society of Chemistry. (**d**) Schematic diagram of the device. (**e**) The EQE and *J_sc_* of the device with and without Mo_1.33_C-doped PEDOT:PSS HTL. (**f**) Schematic diagram of the device. Reproduced with permission from Reference [178]. Copyright 2020, Royal Society of Chemistry. (**g**) The energy band diagrams of employed materials. (**h**) The measured PCE of the device with various HTLs in glove box. Reproduced with permission from Reference [179]. Copyright 2020, Royal Society of Chemistry.

**Figure 20 nanomaterials-11-03170-f020:**
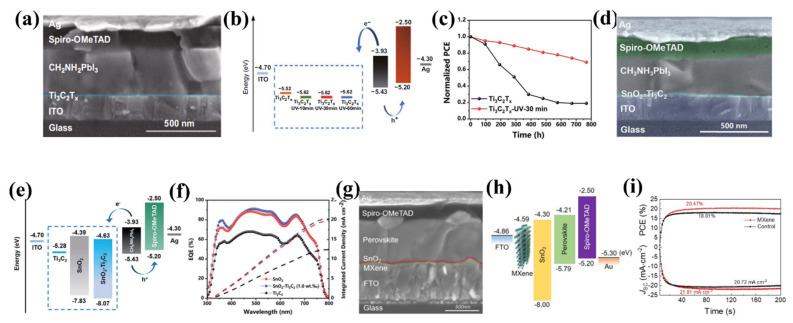
(**a**) The SEM image of the device. (**b**) The energy band diagrams of employed materials. (**c**) The stability measurement of the devices with various ETLs in ambient conditions without encapsulation. Reproduced with permission from Reference [182]. Copyright 2019, Wiley-VCH. (**d**) The SEM image of the device. (**e**) The energy band diagrams of each layer. (**f**) The EQE and *J_sc_* of the device with Ti_3_C_2_, SnO_2_–Ti_3_C_2_ (1.0 wt%), and SnO_2_. Reproduced with permission from Reference [183]. Copyright 2019, Royal Society of Chemistry. (**g**) The SEM image of the device. (**h**) The energy band diagrams of MXene modified PVSK solar cell. (**i**) The PCE and Jsc of the device under their maximum power points. Reproduced with permission from Reference [184]. Copyright 2020, American Chemical Society.

**Figure 21 nanomaterials-11-03170-f021:**
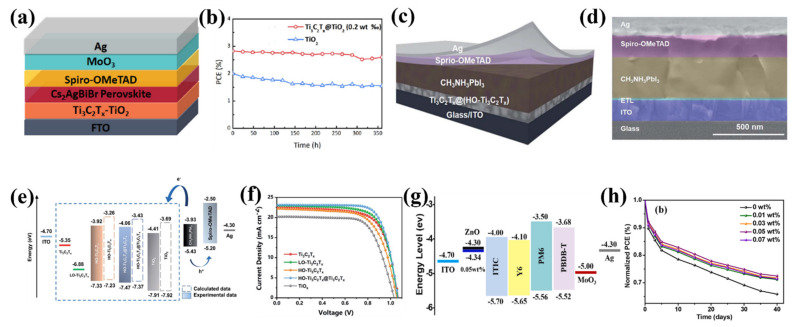
(**a**) Schematic diagram of the device. (**b**) The measured PCE of the device. Reproduced with permission from Reference [185]. Copyright 2021, American Chemical Society. (**c**,**d**) Schematic diagram and SEM image of the device, respectively. (**e**) The energy band diagrams of each layer. (**f**) The *J-V* curve of the device with various ETLs. Reproduced with permission from Reference [186]. Copyright 2021, Royal Society of Chemistry. (**g**) The energy band diagrams of employed materials. (**h**) The stability values without the encapsulation under ambient conditions. Reproduced with permission from Reference [187]. Copyright 2021, Elsevier B.V.

**Figure 22 nanomaterials-11-03170-f022:**
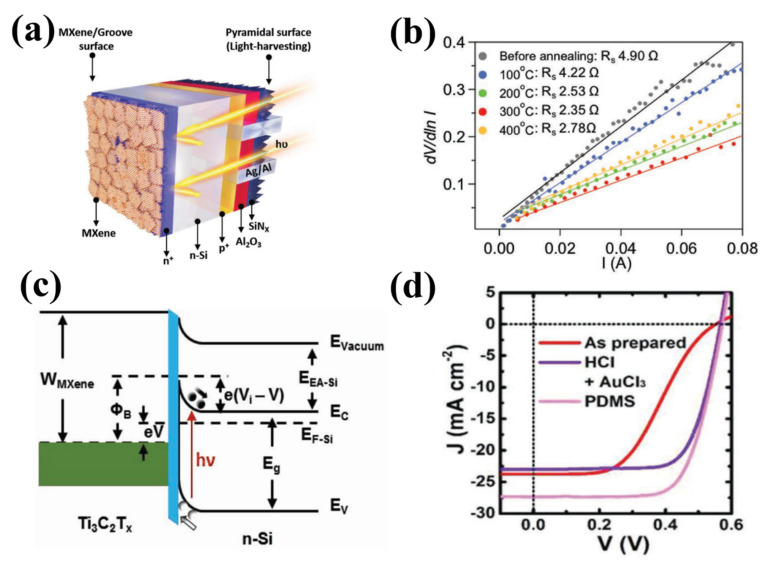
(**a**) Schematic diagram of the device. (**b**) The series resistance (*R_s_*) of the device extrapolated from d_V_/d_ln_ versus curves with and without RTA treatment. Reproduced with permission from Reference [188]. Copyright 2019, Wiley-VCH. (**c**) The energy band diagrams of Ti_3_C_2_T_x_/SiO_2_/Si heterojunction. (**d**) The *J-V* curve of the device under illumination. Reproduced with permission from Reference [189]. Copyright 2019, Wiley-VCH.

## Data Availability

Not applicable.

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
