# Peer review of "MXene-Based Materials for Solar Cell Applications"

_nanomaterials, 2021, doi:10.3390/nano11123170_

Round 1
Reviewer 1 Report
Shi et al. prepared a detailed review about the application of MXenes in photovoltaic devices. The manuscript contains compiled information about basic structural, surface, and semiconductor properties of MXenes and motivation for the use in the structures of the solar cells.
Despite the high quality of the original manuscript, there are several major issues, which require to re-shape the structure of the review or to improve the logical connections between the paragraphs.
Authors open chapter 2 2D materials candidates for solar cell applications with an introduction but then the text is devoted to the absorber material for solar cells – sub-chapter 2.1 Perovskite. The sub-Chapters 2.2 – 2.6 are devoted to other two-dimensional materials, so It’s not clear why the Perovskite chapter is placed at the beginning of the part for other 2D materials.
It’s not clear why the sub-chapter for the description of the basic properties for MXenes was titled 2.6 Other 2D materials… and devoted to only for MXenes.
In chapter 3, the authors wrote “Herein, we summarize and highlight some recent representative investigations 406 regarding the fabrication processes of various solar cells based on 2D materials, including organic, silicon, dye-407 sensitized, and perovskite solar cells.” This point is doubtful because MXenes could improve the performance of the solar cells, could be used as special functional parts of the devices (electrode or transport layer, for example). On the other hand, silicon, dyes, perovskites are not 2D materials.
In chapters, 3.2 – 3.4 author presented technological pathways for the fabrication of the various solar cells. Such a quantity of technical information will reduce the interest during reading. The authors did not formulate the motivation for the use of MXenes to solve the specific problems of each PV technology. Such key points are missed, MXenes have unique properties like the combination of the high conductivity and tunability of the work function in the wide range.
So, each paragraph for the description of the PV technology should be re-shaped. First of all, the review about MXenes does not need such large details about the step-by-step fabrication of solar cells. Authors should formulate the main problems of the PV technologies which could be solved with the use of MXenes.
Other minor notes:
Re-check the labeling for the power of numbers in the text, in several places it’s missed
(for example outstanding electrical properties with high electrical conductivity (≈104 Ω−1 cm−1).
Author Response
Nov. 05, 2021
Dear Editor Alline Wang,
Thank you very much for your great editorial efforts on Nanomaterials and reviewing our manuscript (ID: nanomaterials-1415270). We appreciate the helpful comments and valuable suggestions provided by the reviewers, according to which a highlighted revised version of the manuscript was prepared. A point-by-point response to the reviewers’ comments and the significant changes in the text are included within this letter.
If you could reexamine our manuscript, we would appreciate it very much.
Thank you very much for your consideration.
Best regards!
Sincerely yours,
Zhe Shi
School of Physics & New Energy, Xuzhou University of Technology
Address: Lishui Road NO.2, Xuzhou, Jiangsu, P. R. China, 518060
Tel: +86-15140378104
E-mail: [email protected]
Reviewer 1:
Shi et al. prepared a detailed review about the application of MXenes in photovoltaic devices. The manuscript contains compiled information about basic structural, surface, and semiconductor properties of MXenes and motivation for the use in the structures of the solar cells.
Despite the high quality of the original manuscript, there are several major issues, which require to re-shape the structure of the review or to improve the logical connections between the paragraphs.
Authors open chapter 2 2D materials candidates for solar cell applications with an introduction but then the text is devoted to the absorber material for solar cells – sub-chapter 2.1 Perovskite. The sub-Chapters 2.2 – 2.6 are devoted to other two-dimensional materials, so It’s not clear why the Perovskite chapter is placed at the beginning of the part for other 2D materials.
It’s not clear why the sub-chapter for the description of the basic properties for MXenes was titled 2.6 Other 2D materials… and devoted to only for MXenes.
Response: Thank you very much for your good suggestion. The title of 2 and 2.6 were changed for clarifying the comment.
In chapter 3, the authors wrote “Herein, we summarize and highlight some recent representative investigations 406 regarding the fabrication processes of various solar cells based on 2D materials, including organic, silicon, dye-407 sensitized, and perovskite solar cells.” This point is doubtful because MXenes could improve the performance of the solar cells, could be used as special functional parts of the devices (electrode or transport layer, for example). On the other hand, silicon, dyes, perovskites are not 2D materials.
In chapters, 3.2 – 3.4 author presented technological pathways for the fabrication of the various solar cells. Such a quantity of technical information will reduce the interest during reading. The authors did not formulate the motivation for the use of MXenes to solve the specific problems of each PV technology. Such key points are missed, MXenes have unique properties like the combination of the high conductivity and tunability of the work function in the wide range.
So, each paragraph for the description of the PV technology should be re-shaped. First of all, the review about MXenes does not need such large details about the step-by-step fabrication of solar cells. Authors should formulate the main problems of the PV technologies which could be solved with the use of MXenes.
Response: Thank you very much for your good suggestion. According to your suggestions, after discussion with other authors, we have deleted the chapter 3, meanwhile, the corresponding main problems of the PV technologies which could be solved with the use of MXenes are formulated detailed in chapter 4.
Other minor notes:
Re-check the labeling for the power of numbers in the text, in several places it’s missed (for example outstanding electrical properties with high electrical conductivity (≈104 Ω−1 cm−1).
Response: Thank you very much for your good suggestion. According to your suggestion, we checked out the whole manuscript carefully and these spelling mistakes have been corrected.

Reviewer 2 Report
No particular comments.
Author Response
Nov. 05, 2021
Dear Editor Alline Wang,
Thank you very much for your great editorial efforts on Nanomaterials and reviewing our manuscript (ID: nanomaterials-1415270). We appreciate the helpful comments and valuable suggestions provided by the reviewers, according to which a highlighted revised version of the manuscript was prepared. A point-by-point response to the reviewers’ comments and the significant changes in the text are included within this letter.
If you could reexamine our manuscript, we would appreciate it very much.
Thank you very much for your consideration.
Best regards!
Sincerely yours,
Zhe Shi
School of Physics & New Energy, Xuzhou University of Technology
Address: Lishui Road NO.2, Xuzhou, Jiangsu, P. R. China, 518060
Tel: +86-15140378104
E-mail: [email protected]
Reviewer 1:
Shi et al. prepared a detailed review about the application of MXenes in photovoltaic devices. The manuscript contains compiled information about basic structural, surface, and semiconductor properties of MXenes and motivation for the use in the structures of the solar cells.
Despite the high quality of the original manuscript, there are several major issues, which require to re-shape the structure of the review or to improve the logical connections between the paragraphs.
Authors open chapter 2 2D materials candidates for solar cell applications with an introduction but then the text is devoted to the absorber material for solar cells – sub-chapter 2.1 Perovskite. The sub-Chapters 2.2 – 2.6 are devoted to other two-dimensional materials, so It’s not clear why the Perovskite chapter is placed at the beginning of the part for other 2D materials.
It’s not clear why the sub-chapter for the description of the basic properties for MXenes was titled 2.6 Other 2D materials… and devoted to only for MXenes.
Response: Thank you very much for your good suggestion. The title of 2 and 2.6 were changed for clarifying the comment.
In chapter 3, the authors wrote “Herein, we summarize and highlight some recent representative investigations 406 regarding the fabrication processes of various solar cells based on 2D materials, including organic, silicon, dye-407 sensitized, and perovskite solar cells.” This point is doubtful because MXenes could improve the performance of the solar cells, could be used as special functional parts of the devices (electrode or transport layer, for example). On the other hand, silicon, dyes, perovskites are not 2D materials.
In chapters, 3.2 – 3.4 author presented technological pathways for the fabrication of the various solar cells. Such a quantity of technical information will reduce the interest during reading. The authors did not formulate the motivation for the use of MXenes to solve the specific problems of each PV technology. Such key points are missed, MXenes have unique properties like the combination of the high conductivity and tunability of the work function in the wide range.
So, each paragraph for the description of the PV technology should be re-shaped. First of all, the review about MXenes does not need such large details about the step-by-step fabrication of solar cells. Authors should formulate the main problems of the PV technologies which could be solved with the use of MXenes.
Response: Thank you very much for your good suggestion. According to your suggestions, after discussion with other authors, we have deleted the chapter 3, meanwhile, the corresponding main problems of the PV technologies which could be solved with the use of MXenes are formulated detailed in chapter 4.
Other minor notes:
Re-check the labeling for the power of numbers in the text, in several places it’s missed (for example outstanding electrical properties with high electrical conductivity (≈104 Ω−1 cm−1).
Response: Thank you very much for your good suggestion. According to your suggestion, we checked out the whole manuscript carefully and these spelling mistakes have been corrected.
Reviewer 3:
In this manuscript, the author reports, ‘MXene-based materials for solar cell applications’. The current study is on a topic of relevance and general interest to readers in this area. The authors should address the following questions before getting a possible publication. Recommendation: Major revisions needed as noted.
- The novelty of the review should be discussed in the Introduction section.
Response: Thank you very much for your good suggestion. According to your suggestion, the introduction section has been re-wrote.
- Figures resolution should be increased.
Response: Thank you very much for your good suggestion. All figures cited in this paper have been re-edited to improve the resolution.
- The formatting and grammatical errors in the article need to be checked carefully.
Response: Thank you very much for your good suggestion. The English writing of this manuscript has been examined carefully and the grammatical errors have been corrected.
- The section 4 title should be modified. Synthesis and property of which materials that should be mentioned.
Response: Thank you very much for your good suggestion. The title is modified as follows: Synthesis and property of 2D MXenes
- The texts in several Figures are not clearly visible. The author should take care of that. Some examples are Fig. 10,11, 12 19, 20, 21, 22, 23
Response: Thank you very much for your good suggestion. All figures cited in this paper have been re-edited to improve the resolution, in particular the figures mentioned above.
- The authors have cited relevant references in the Introduction section; however the topic background of the manuscript needs to be highlighted further to broaden the impact, related literatures: Small, 17(32), 2101925; ACS Applied Materials & Interfaces, 13(26), 31038-31050; MXenes for solar cells. Nano-Micro Letters, 13(1), 1-17; Polymer Bulletin, 77(6), 2923-2943.
Response: Thank you very much for your good suggestion. The introduction section has been re-wrote based on these related literatures.
Reviewer 3 Report
In this manuscript, the author reports, ‘MXene-based materials for solar cell applications’. The current study is on a topic of relevance and general interest to readers in this area. The authors should address the following questions before getting a possible publication. Recommendation: Major revisions needed as noted. 1. The authors described some experiment methods in this manuscript to illustrate how to synthesis nanomaterials. Sometimes, figure or table may help authors clearly display those methods. 2. The novelty of the review should be discussed in the Introduction section. 3. Figures resolution should be increased. 4. The formatting and grammatical errors in the article need to be checked carefully. 5. The section 4 title should be modified. Synthesis and property of which materials that should be mentioned. 6. The texts in several Figures are not clearly visible. The author should take care of that. Some examples are Fig. 10,11, 12 19, 20, 21, 22, 23 7. The authors have cited relevant references in the Introduction section; however the topic background of the manuscript needs to be highlighted further to broaden the impact, related literatures: Small, 17(32), 2101925; ACS Applied Materials & Interfaces, 13(26), 31038-31050; MXenes for solar cells. Nano-Micro Letters, 13(1), 1-17; Polymer Bulletin, 77(6), 2923-2943
Author Response
Nov. 05, 2021
Dear Editor Alline Wang,
Thank you very much for your great editorial efforts on Nanomaterials and reviewing our manuscript (ID: nanomaterials-1415270). We appreciate the helpful comments and valuable suggestions provided by the reviewers, according to which a highlighted revised version of the manuscript was prepared. A point-by-point response to the reviewers’ comments and the significant changes in the text are included within this letter.
If you could reexamine our manuscript, we would appreciate it very much.
Thank you very much for your consideration.
Best regards!
Sincerely yours,
Zhe Shi
School of Physics & New Energy, Xuzhou University of Technology
Address: Lishui Road NO.2, Xuzhou, Jiangsu, P. R. China, 518060
Tel: +86-15140378104
E-mail: [email protected]
Reviewer 3:
In this manuscript, the author reports, ‘MXene-based materials for solar cell applications’. The current study is on a topic of relevance and general interest to readers in this area. The authors should address the following questions before getting a possible publication. Recommendation: Major revisions needed as noted.
- The novelty of the review should be discussed in the Introduction section.
Response: Thank you very much for your good suggestion. According to your suggestion, the introduction section has been re-wrote.
- Figures resolution should be increased.
Response: Thank you very much for your good suggestion. All figures cited in this paper have been re-edited to improve the resolution.
- The formatting and grammatical errors in the article need to be checked carefully.
Response: Thank you very much for your good suggestion. The English writing of this manuscript has been examined carefully and the grammatical errors have been corrected.
- The section 4 title should be modified. Synthesis and property of which materials that should be mentioned.
Response: Thank you very much for your good suggestion. The title is modified as follows: Synthesis and property of 2D MXenes
- The texts in several Figures are not clearly visible. The author should take care of that. Some examples are Fig. 10,11, 12 19, 20, 21, 22, 23
Response: Thank you very much for your good suggestion. All figures cited in this paper have been re-edited to improve the resolution, in particular the figures mentioned above.
- The authors have cited relevant references in the Introduction section; however the topic background of the manuscript needs to be highlighted further to broaden the impact, related literatures: Small, 17(32), 2101925; ACS Applied Materials & Interfaces, 13(26), 31038-31050; MXenes for solar cells. Nano-Micro Letters, 13(1), 1-17; Polymer Bulletin, 77(6), 2923-2943.
Response: Thank you very much for your good suggestion. The introduction section has been re-wrote based on these related literatures.

Round 2
Reviewer 1 Report
The authors performed a great job on the revision.
However, the logical sequence of the review still requires minor revision.
The authors completely deleted the part with a brief description of the OPV and Si solar cells. On the other nad several examples for the use of MXenes for charge-transporting layers and electrodes were done specially for these type of PV devices. Thus, I will ask the authors to put the part with a BRIEF description of OPV and Silicon solar cells architectures in the manuscript (from the deleted parts).
Other minors:
Page 2 line 65, please check do you mean the value for conductivity 104 or 10000?
Please, perform the spell check
Author Response
Nov. 13, 2021
Dear Editor Alline Wang,
Thank you very much for your great editorial efforts on Nanomaterials and reviewing our manuscript (ID: nanomaterials-1415270). We appreciate the helpful comments and valuable suggestions provided by the reviewers, according to which a highlighted revised version of the manuscript was prepared. A point-by-point response to the reviewers’ comments and the significant changes in the text are included within this letter.
If you could reexamine our manuscript, we would appreciate it very much.
Thank you very much for your consideration.
Best regards!
Sincerely yours,
Zhe Shi
School of Physics & New Energy, Xuzhou University of Technology
Address: Lishui Road NO.2, Xuzhou, Jiangsu, P. R. China, 518060
Tel: +86-15140378104
E-mail: [email protected]
Reviewer1:
The authors performed a great job on the revision.
However, the logical sequence of the review still requires minor revision.
The authors completely deleted the part with a brief description of the OPV and Si solar cells. On the other nad several examples for the use of MXenes for charge-transporting layers and electrodes were done specially for these type of PV devices. Thus, I will ask the authors to put the part with a BRIEF description of OPV and Silicon solar cells architectures in the manuscript (from the deleted parts).
Response: Thanks very much for your good suggestion. According to your suggestion, we add a brief description of OPV and Silicon solar cells architectures in the revised manuscript and marked as blue color.
Other minors:
Page 2 line 65, please check do you mean the value for conductivity 104 or 10000?
Please, perform the spell check
Response: Thanks very much for your good suggestion. This value should be 104 Ω−1 cm−1 instead of 104 Ω−1 cm−1. It has been corrected and marked as yellow color in the revised manuscript.
Reviewer 3 Report
The authors have addressed all the queries raised before. Therefore the manuscript can be accepted in the present form.
Author Response

(The authors gave the same response as above.)
